# A minority of final stacks yields superior amplitude in single-particle cryo-EM

Jianying Zhu[1,10], Qi Zhang[2,3,4,5,10], Hui Zhang[6], Zuoqiang Shi[1,7] ✉,
Mingxu Hu [2,3,4,5,8] ✉ & Chenglong Bao [1,7,9] ✉

Cryogenic electron microscopy (cryo-EM) is widely used to determine near-atomic resolution structures of biological macromolecules. Due to the low signal-to-noise ratio, cryo-EM relies on averaging many images. However, a crucial question in the field of cryo-EM remains unanswered: how close can we get to the minimum number of particles required to reach a specific resolution in practice? The absence of an answer to this question has impeded progress in understanding sample behavior and the performance of sample preparation methods. To address this issue, we develop an iterative particle sorting and/or sieving method called CryoSieve. Extensive experiments demonstrate that CryoSieve outperforms other cryo-EM particle sorting algorithms, revealing that most particles are unnecessary in final stacks. The minority of particles remaining in the final stacks yield superior high-resolution amplitude in reconstructed density maps. For some datasets, the size of the finest subset approaches the theoretical limit.

The transformative impact of cryo-EM single-particle analysis (SPA) on the field of structural biology has been widely recognized by the scientific community[1]. Cryo-EM has advanced significantly due to a series of technological innovations[2–7], enabling the technique to provide macromolecular structures with up to atomic resolution at an unprecedented rate. This technological progress is commonly referred to as the resolution revolution[8]. Cryo-EM involves using electron microscopy images of biomolecules embedded in vitreous, glass-like ice[9], which are then combined to generate three-dimensional density maps. These maps provide valuable insights into the function of macromolecules and their role in biological processes.

The stability and electron-optical performance of electron microscopes do not hinder the use of cryo-EM[10]. However, biological samples studied in cryo-EM are radiation-sensitive[11,12]. Therefore, a trade-off must be made between improving the signal-to-noise ratio (SNR) and limiting radiation damage[13,14]. It was concluded that statistically well-defined three-dimensional (3D) structures could not be

obtained from individual biological macromolecules at atomic resolution[15,16]. Instead, increasing the SNR by averaging image data from many identical macromolecules is the only way to progress[13,17,18]. Over two decades ago, Henderson estimated that structures could be determined at a resolution of nearly 3 Å by merging data from approximately 12,000 particles, even for particles as small as approximately 40 kDa[19]. Later, Rosenthal and Henderson argued that the electron microscopy community should adopt the same threshold criterion for structure factor quality as the X-ray protein crystallography community, which was set at a figure-of-merit of 0.5 corresponding to a phase error of 60°[16]. The theoretical limit of the minimum number of particle images required to achieve a specific resolution can be calculated using the theory proposed by Henderson and Rosenthal[16,19], given the B-factor of the instrument (e.g., electron microscopy and camera)[13,14,20]. In practice, the final stacks of cryo-EM still far fall short of the theoretical limit, indicating a considerable gap between what can be accomplished and the

[1]Yau Mathematical Sciences Center, Tsinghua University, Beijing, China. [2]Key Laboratory of Protein Sciences (Tsinghua University), Ministry of Education, Beijing, China. [3]School of Life Science, Tsinghua University, Beijing, China. [4]Beijing Advanced Innovation Center for Structural Biology, Beijing, China. [5]Beijing Frontier Research Center for Biological Structure, Beijing, China. [6]Qiuzhen College, Tsinghua University, Beijing, China. [7]Yanqi Lake Beijing Institute of Mathematical Sciences and Applications, Beijing, China. [8]Shenzhen Academy of Research and Translation, Shenzhen, China. [9]State Key Laboratory of Membrane Biology, School of Life Sciences, Tsinghua University, Beijing, China. [10]These authors contributed equally: Jianying Zhu, Qi Zhang. ✉e-mail: zqshi@tsinghua.edu.cn; humingxu@smart.org.cn; clbao@tsinghua.edu.cn

physical limit of what cryo-EM can do[21]. The initial particle datasets obtained by particle picking from micrographs undergo multiple rounds of laborious 2D and 3D classification to generate the final stack for model determination. The final stacks, which yield atomic or sub-atomic resolution density maps, typically comprise several orders of magnitude fewer particles than the original datasets. Therefore, the cryo-EM field faces the long-standing question of how close we can approach the theoretical limit in practice. The lack of an answer to this open question has hindered the quantification of the performance of various underdeveloped sample preparation methods and impeded the investigation of trends and the understanding of the underlying mechanisms of sample behavior. To answer the question of how close cryo-EM can approach its theoretical limit, it is crucial to determine the minimum number of particles required to achieve a high-resolution 3D reconstruction within a given dataset.

In this work, we introduce CryoSieve[22], an iterative particle sorting and/or sieving algorithm that identifies the smallest subset of particles necessary to generate high-resolution density maps, which we call the finest subset. CryoSieve compares the high-frequency components of synthetic and observed particle images. A higher CryoSieve score indicates superior quality rather than typical cryo-EM damage or artifacts. Extensive experiments show that CryoSieve outperforms other particle sorting algorithms in various metrics and reveal that most particles in final stacks are futile. The finest subsets generate 3D density maps with better high-resolution amplitude, using much fewer particles than the final stacks. We propose that CryoSieve removes radiation-damaged particles within cryo-EM datasets, supported by experiments on the dataset consisting of particles exposed to various levels of electron dose. Finally, we compare the minimum particles required in theory with the size of the finest subsets obtained by CryoSieve, finding that some datasets come close to the theoretical limit after being sieved by CryoSieve. From our experiments, we suggest that advancements during the sample preparation process, aimed at increasing the proportion of the finest subset in the final stack, could potentially facilitate the development of cryo-EM.

## Results

### Design of CryoSieve

We have developed a particle sorting and/or sieving model called CryoSieve that iteratively performs 3D reconstruction and particle selection, eliminating futile particles during each iteration. A flow chart scheme is provided in Supplementary Fig. 3. In CryoSieve, the relion_reconstruct module of RELION is used to reconstruct a new density map with the retained particle images, which is then used in the subsequent iteration. The retained particle images in each iteration form a subset of those from the previous iteration, as shown in the following formula:

$$\left\{ i_1^{(k)}, i_2^{(k)} \cdots, i_{n^{(k)}}^{(k)} \right\} \subset \left\{ i_1^{(k-1)}, i_2^{(k-1)}, \cdots, i_{n^{(k-1)}}^{(k-1)} \right\}, \quad (1)$$

where $n^{(k-1)}$ represents the number of retained particles. At each iteration, let $b_j$ be the $j$-th particle image, $A_j$ be its forward operator defined by the estimated parameters and $x^{(k-1)}$ be the reconstructed density map from the retained particle images in the previous iteration, particles are sieved out based on their CryoSieve score, which is defined as follows:

$$g_j := \left\| H^{(k)} b_j \right\|_2^2 - \left\| H^{(k)} \left( b_j - A_j x^{(k-1)} \right) \right\|_2^2, j \in \left\{ i_1^{(k-1)}, i_2^{(k-1)}, \cdots, i_{n^{(k-1)}}^{(k-1)} \right\}. \quad (2)$$

Here, $H^{(k)}$ is the highpass operator at the $k$-th iteration, and its threshold frequency increases linearly as the iteration progresses (Supplementary Table 4). Given that $g_j$ relies on the accurate amplitude of the reconstructed density map $x^{(k)}$, CryoSPARC is not the optimal choice for reconstruction in the particle selection step (Supplementary Fig. 2). It tends to deviate significantly from the true amplitude (Supplementary Fig. 2c). Furthermore, the amplitude information within the CryoSieve score proves vital, and the phase residual is ineffective as a metric for particle selection (Supplementary Fig. 4).

The CryoSieve score estimates the similarity between a particle and a reference projection above a given frequency. A higher CryoSieve score indicates that the particle and the reference projection share a higher proportion of signal energy, indicating better particle quality. As radiation damage mainly affects the high-frequency range, the CryoSieve score includes a highpass operator to extract the high-frequency part. We have demonstrated that the CryoSieve score can identify particles with incorrect pose parameters or components in the high-frequency range through theoretical analysis and simulation verification (Supplementary Material I and III). Specifically, assuming that noise in particles follows a Gaussian distribution, we have shown that, with high probability, the CryoSieve score is an ideal indicator of particle image quality, distinguishing it from typical cryo-EM damage or artifacts (Supplementary Material I). Furthermore, the CryoSieve score exhibits remarkable accuracy in removing particles with incorrect pose and CTF parameter estimations, achieving a high accuracy of over 90% (Supplementary Material III).

### Majority of the particles are futile in final stacks

We demonstrate the versatility of our method by applying it to eight experimental datasets (Table 1). The first dataset is derived from the human TRPA1 ion channel (EMPIAR-10024)[23]. The second dataset is from influenza hemagglutinin trimer (EMPIAR-10097)[24], of which the preferred orientation necessitated 40° tilts during data acquisition. The third dataset involves LAT1-CD98hc bound to MEM-108 Fab (EMPIAR-10264)[25], while the fourth features membrane-bound pfCRT complexed with Fab (EMPIAR-10330)[26]. Both of these datasets utilized signal subtraction during data processing. The fifth dataset is from CS-17 Fab-bound TSHR-Gs (EMPIAR-11120)[27]. The sixth is from TRPM8 bound to calcium (EMPIAR-11233)[28]. The seventh dataset is derived from human apoferritin (EMPIAR-10200)[29], achievable to a resolution

**Table 1 | Microscopic imaging parameters of eight experimental datasets along with their associated metadata**

| Dataset | TEM | Electron detector | Number of particles | Spherical aberration (mm) | Symmetry | Molecular weight (kDa) |
|---|---|---|---|---|---|---|
| TRPA1 | TF30 Polara | Gatan K2 Summit | 43,585 | 2.0 | $C_4$ | 688 |
| hemagglutinin | Titan Krios | Gatan K2 Summit | 130,000 | 2.7 | $C_3$ | 150 |
| LAT1 | Titan Krios | FEI FALCON III | 250,712 | 2.7 | $C_1$ | 172 |
| pfCRT | Titan Krios | Gatan K2 Qutuamn | 16,905 | 0.001 | $C_1$ | 102 |
| TSHR-Gs | Titan Krios | Gatan K3 Qutuamn | 41,054 | 2.7 | $C_1$ | 125 |
| TRPM8 | Titan Krios | Gatan K2 Summit | 42,040 | 2.6 | $C_4$ | 513 |
| apoferritin | Titan Krios | Gatan K2 Summit | 382,391 | 2.7 | $O$ | 440 |
| streptavidin | Titan Krios | Gatan K2 Summit | 23,991 | 0.01 | $D_2$ | 52 |

above 2Å. The eighth dataset originates from streptavidin (EMPIAR-10269)[30], with a molecular weight of only 52 kDa. All datasets were obtained using a voltage of 300 kV and an amplitude contrast of 0.07 or 0.1. The TEM systems and electron detectors used in the experiments are listed in Table 1, along with additional metadata such as the number of particles in the final stacks, spherical aberration, symmetry and molecular weight.

All of the datasets are deposited in the Electron Microscopy Public Image Archive (EMPIAR) [31] as final stacks. These final stacks, which also contain the corresponding refined Euler angles, were used to generate the final published reconstructions. The final stacks are generated by manually selecting significantly smaller subsets through multiple rounds of 2D/3D classification, resulting in a substantially reduced number of particles compared to the original particle stacks.

We employed CryoSieve to process the eight experimental datasets. CryoSieve removed 20% of the particles in each iteration, resulting in a retaining ratio of 80.0%, 64.0%, 51.2%, and so on. The highpass cutoff frequency of CryoSieve increases linearly across iterations. The retained particles in different iterations were then used for ab initio reconstruction to determine the finest subset of particles. The finest subset only contained 21.0% to 32.8% of the particles in the final stack. However, the quality of the reconstructed map from the finest subset was consistent with that obtained from all particles in the final stack, as demonstrated in Fig. 1. For some datasets, the density maps showed a certain degree of improvement, which was visualized by the restoration of some previously blurred or missing side chains in the density map (Supplementary Fig. 8). The results demonstrate that CryoSieve is proficient in discarding more than half of the particles, utilizing the CryoSieve score—a metric reflecting the discrepancy between the particle image and its reference projection. Crucially, this process does not compromise the quality of the final reconstruction. Moreover, the pose distribution of the removed particles was similar to those of all particles in the final stacks (Supplementary Fig. 6). Therefore, CryoSieve is highly effective in selecting the most informative particles.

We performed a comparative analysis of CryoSieve with other cryo-EM particle sorting criteria or software currently used in the field, including the normalized cross-correlation (NCC) method[32], the angular graph consistency (AGC) approach[33] and the non-alignment classification[6]. The parameter settings for CryoSieve and the other comparative algorithms were listed in Supplementary Material VI. In our experiments, we used final stacks composed of relatively high-quality particles. NCC retains an equal number of particles compared to CryoSieve at each iteration, while AGC's retaining ratio is self-determined. However, AGC's retaining ratio was mainly over 90%, resulting in only a small fraction of particles being removed. Thus, the quality of the reconstructed map using the retained particles did not improve or worsen (Supplementary Table 2), as these tested final stacks are composed of relatively high-quality particles. For the non-alignment classification applied to hemagglutinin, LAT1, and apo-ferritin, less than half of the particles were removed, resulting in some enhancement (Supplementary Material V). However, this enhancement still falls notably short of the results achieved by CryoSieve (Supplementary Material V). For the other five datasets, the retaining ratios using non-alignment classification exceeded 90%, resulting in the quality of maps reconstructed from the retained particles either remaining unchanged or deteriorated (Supplementary Material V). Additionally, we randomly selected the same number of particles from the tested final stacks at each iteration to observe the baseline effect of particle number reduction.

For all the aforementioned methods (CryoSieve, NCC, AGC, non-alignment classification, and random), we discarded the refined Euler angles published and deposited on EMPIAR to prevent the inadvertent transfer of information from the removed particles to the retained particles. Thus, the retained particles were used for ab initio

reconstruction by CryoSPARC to obtain refreshed sets of Euler angles and density maps. Several metrics, including FSC-based resolution[16], Q-score[34] and Rosenthal-Henderson B-factor[16] were used to measure the quality of the refreshed density maps. Based on these metrics, our analyses reveal that CryoSieve effectively sieves out 67.2% to 79.0% (varying based on datasets) of particles from the final stacks without deteriorating the yielded density maps (Fig. 2). In contrast, subsets of equal size retained by the other methods failed to reconstruct density maps of the same quality as the original (Fig. 2). Therefore, CryoSieve significantly outperforms other particle sorting algorithms, demonstrating that the majority of particles are dispensable in the final stacks. A key factor in CryoSieve's superiority over both NCC, AGC and non-alignment classification is the integration of the highpass operator when computing the CryoSieve score. Without the truncation of high frequencies, scores may be predominantly influenced by low-frequency components, making it challenging to differentiate non-contributory particles in cryo-EM.

CisTEM[5] can report a score for each single-particle image after 3D refinement. During the 3D refinement process of cisTEM, the pose parameters of particles are re-estimated or refined. Therefore, due to differences in alignment and other image processing workflows between cisTEM and CryoSPARC, cisTEM cannot be strictly compared with CryoSieve. We compared CryoSieve and cisTEM by sorting particles using the cisTEM score and retaining equal particle counts for ab initio reconstruction in CryoSPARC (details in Supplementary Material II). CryoSieve outperformed cisTEM in all eight experimental datasets (Fig. 2).

We analyzed the differences between the particle images retained and removed using CryoSieve by performing 2D classification of the particles into 50 classes using CryoSPARC. To ensure a comparable number of particles for both retained and removed groups, we ran CryoSieve and terminated at the third iteration, yielding a retention ratio of 51.2% and a removal ratio of 48.8%. CryoSPARC reported the 2D resolution of each class, along with the number of particle images belonging to it. The particles retained by CryoSieve (Fig. 3, steel blue) were distributed at a higher resolution compared to those removed by it (Fig. 3, crimson). In six out of the eight datasets, particle images with the highest resolution, i.e., 7.4–7.1 Å in TRPA1, 8.5–9.6 Å in hemagglutinin, 6.6–8.2 Å in LAT1, 7.2–11.6 Å in pfCRT, 7.2–8.5 Å in TSGH-Gs, and 11.6–7.5 Å in TRPM8, were entirely retained by CryoSieve. For apo-ferritin, the majority of particles within the highest resolution range (5.5–5.3 Å) were constituted by the particles retained by CryoSieve. However, for streptavidin, possibly due to the adoption of a phase plate during data collection, unusually high resolutions were reported in the 2D classification step, rendering such a comparison between retained and removed particles ineffective. In conclusion, our analysis suggests that CryoSieve selectively retained the higher-quality particle images in the final stack while discarding lower-quality ones. It is noted that some information remains in these discarded particles, but it does not enhance the information present in the finest subset (Supplementary Material VII).

## Better high-resolution amplitude with much fewer particles
B-factors, also known as Debye-Waller factors or temperature factors, reflect the rate at which the amplitude of high-resolution information decreases[16]. Lower B-factors indicate that the high-resolution signal has been better preserved during sample preparation, imaging, and image processing, implying that the particle images are of higher quality. B-factors are widely used to measure image quality in cryo-EM quantitatively[35–39]. In our eight experimental datasets, the finest subset, consisting of only 21.0% to 32.8% of particles in the final stack, generates 3D density maps with the Rosenthal and Henderson's B-factors reduced by 21.1 Å$^2$ to 169.0 Å$^2$, in comparison to those produced by the original final stacks (Table 2, column D and E). The process of fitting and solving for Rosenthal and Henderson's B-factors is visualized in

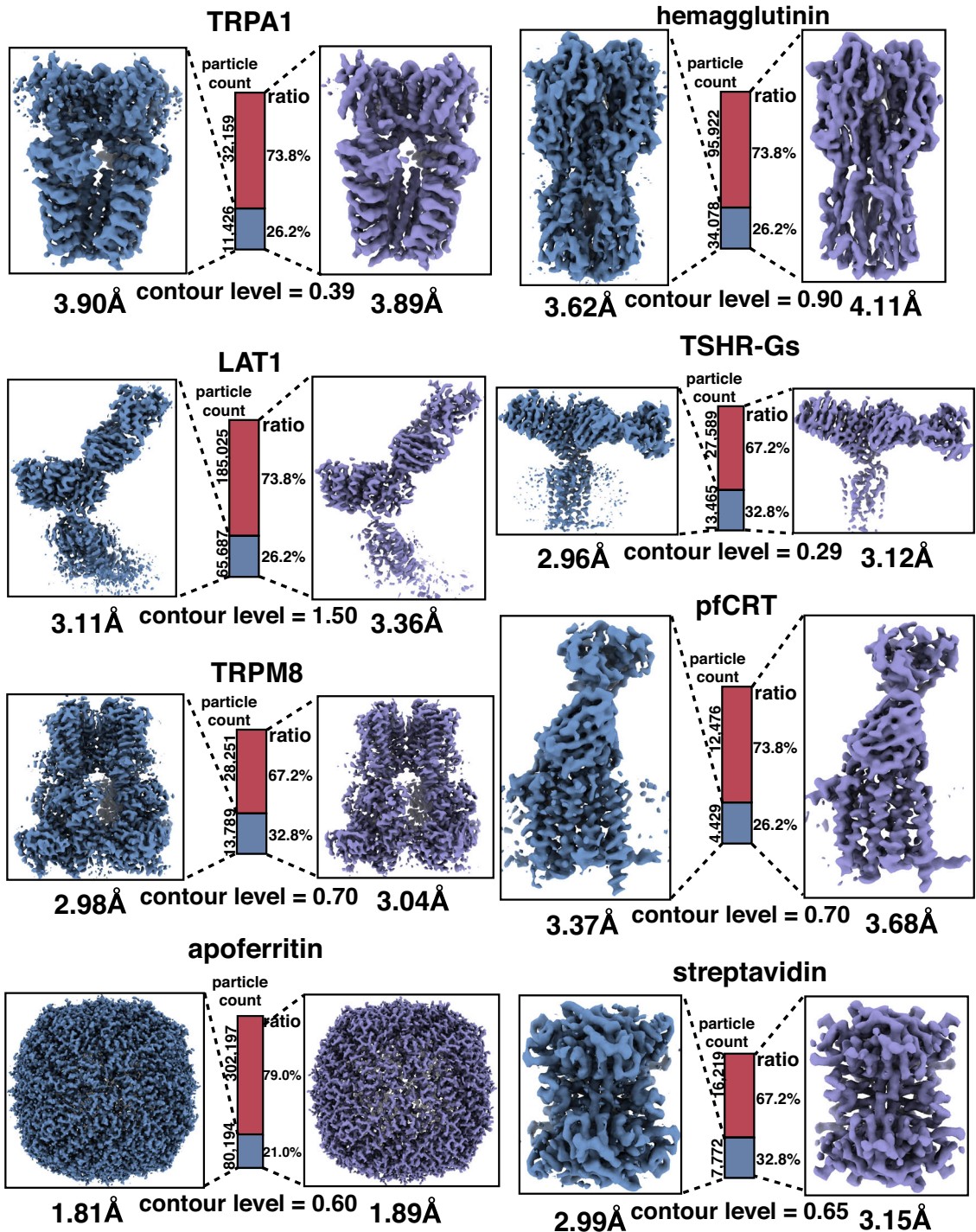

**Fig. 1 | CryoSieve is capable of maintaining resolutions after removing the majority of particles in the final stacks.** For all eight experimental datasets, density maps of the CryoSieve-retained particles (steel blue) and all particles in the final stack (medium purple) were compared, obtained from CryoSPARC's ab initio reconstruction after discarding the published refined Euler angles deposited on EMPIAR to avoid the bias in the final stack. The density maps were first FSC-weighted (based on FSCs given by CryoSPARC), and then B-factor sharpened using

equivalent B-factors for the same protein: $-90$ Å$^2$ for TRPA1, $-180$ Å$^2$ for hemagglutinin, $-100$ Å$^2$ for LAT1, $-60$ Å$^2$ for pfCRT, $-70$ Å$^2$ for TSHR-Gs, $-80$ Å$^2$ for TRPM8, $-65$ Å$^2$ for apoferritin, and $-110$ Å$^2$ for streptavidin. The central bars indicate the proportions of the retained and removed particles. The equivalent contour level was applied for each protein respectively, as indicated at the base of each ratio bar. The raw density maps corresponding to these results, unsharpened by B-factor, are given in Supplementary Fig. 1.

Supplementary Fig. 5. Moreover, the B-factors determined by CryoSPARC are presented in Table 2, columns B and C. In other words, the density maps reconstructed from the finest subset have a better high-resolution amplitude, meaning they contain a greater high-resolution

intensity, despite the fact that the finest subsets only contain a small fraction of particles in the final stack. This indicates that CryoSieve significantly reduced the temperature factor and alleviated the amplitude contrast decay, suggesting that high-quality particles

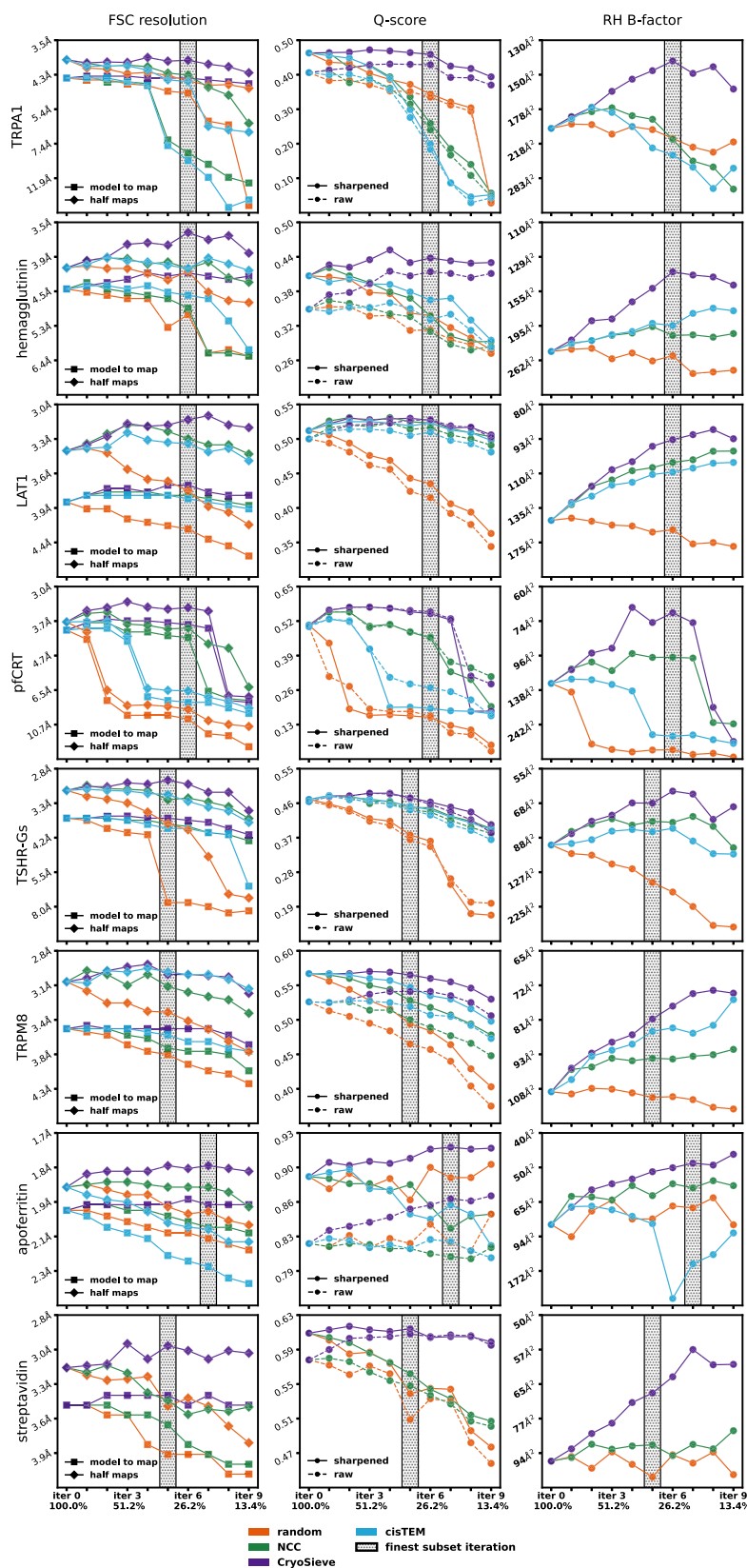

**Nature Communications** | (2023)14:7822

contribute to the density map and can be effectively selected by CryoSieve.

**CryoSieve can effectively detect radiation-damaged particles**

We hypothesize that some particle images in the final stacks have been subject to some degree of radiation damage and cannot be screened out by conventional methods. These particles do not contribute positively to the reconstructed density map. To verify the possibility of this conjecture, we acquired micrograph movie stacks of the proteasome using a Titan Krios 300 keV cryo-EM equipped with a K3 direct electron detection camera. The defocus range was set between 0.5 μm and 1.5 μm. Each stack comprised 32 frames with a total electron dose

**Fig. 2 | CryoSieve outperformed other algorithms in terms of FSC-based resolutions, Q-scores and Rosenthal-Henderson B-factors.** We compared the density maps reconstructed from retained particles obtained by CryoSieve (indigo), NCC (green) and cisTEM (light blue), along with random (orange) as the baseline, at different retention ratios. Density maps were ab initio reconstructed by CryoSPARC after discarding the published refined Euler angles deposited on EMPIAR. Five metrics evaluated density map quality. The first column presents FSC-based resolutions: model-to-map (solid lines with squares) and two-half-maps (solid lines with diamonds). The second column shows Q-scores for raw (dashed lines with circles) and sharpened maps (solid lines with circles), with sharpening B-factors determined by CryoSPARC. The third column depicts Rosenthal-Henderson B-factors. The iterations where CryoSieve obtained the finest subset, determined by comprehending these metrics, are labeled with hatched bars. Due to the involvement of the phase plate in the streptavidin dataset, cisTEM failed to refine the poses, and thus the corresponding analysis was omitted.

of 50 $e^-Å^{-2}$. The electron dose was uniformly distributed across all frames. Particles were picked from identical positions using averages from frames 5–14, 10–19, 15–24, and 20–29. Consequently, we constructed a dataset consisting of 183,464 particles that represented four different levels of absorbed electron doses (Fig. 4a).

We assessed the retention behavior of CryoSieve, NCC, cisTEM, AGC, and non-alignment classification in particles subjected to varying radiation damage levels, using random retention as a comparative baseline. As the number of iterations increased, the retention rate diminished. Notably, CryoSieve demonstrated enhanced proficiency in identifying particles with elevated radiation damage levels relative to NCC and cisTEM (Fig. 4b). The retention ratio for cisTEM was equated to each iteration of CryoSieve (Supplementary Material II). For AGC and non-alignment classification, the retention ratio was autonomously determined. We simultaneously compared the distribution of particles across the four radiation damage levels, selecting the sixth iteration (with a retention ratio of 26.2%) of CryoSieve, NCC, and cisTEM for this analysis. The analysis also incorporated particles retained by the AGC and non-alignment classification methods, with retention ratios auto-determined for these methods (Fig. 4c). Model-to-map FSCs (Fig. 4d) and a thorough comparison of density maps (Fig. 4e) affirmed CryoSieve's superiority over the other methods. Retained particles, utilizing the cisTEM score as a selection criterion, exhibited a preferred orientation, resulting in diminished quality (Supplementary Fig. 7).

While the approach of grouping frames from micrograph movie stacks cannot remove other potential complications that particles might endure, such as erroneous poses, CTF parameters, and denaturation, we sought additional validation. To this end, we employed InSilicoTEM to generate synthesized particles exhibiting varying simulated radiation damage. With these simulated radiation-damage datasets, CryoSieve consistently outperformed all other methods. Notably, in the final iterations, CryoSieve exclusively retained particles unaffected by radiation damage (Supplementary Material VIII).

It is worth noting that CryoSieve can efficiently remove particles with incorrect pose and CTF parameter estimations, achieving a high accuracy of over 90% (Supplementary Material III). However, these particles are also removed by the non-alignment classification approach (Supplementary Material III), making them unlikely to be present in the final stacks.

### The finest subsets may be close to the theoretical number of particles limit

The theoretical number limit of particle images, given by Rosenthal and Henderson[16], is

$$N_{particles} = \frac{1}{N_{asym}} \frac{\frac{S^2}{N^2} 30\pi}{N_e \sigma_e d} \exp\left(\frac{B}{2d^2}\right), \quad (3)$$

where $N_{asym}$, $\frac{S}{N}$, $N_e$, $\sigma_e$, $d$, $B$ stand for the number of asymmetric units, the signal-to-noise threshold criteria of the resolution, the electron dose, the elastic cross-section for carbon, the resolution, and the overall temperature factor, respectively. In the above formula, $\frac{S}{N} = \frac{1}{\sqrt{3}}$, which is equivalent to a phase error of 60° or 0.143-threshold of half-maps FSC[16]. Meanwhile, $N_e = 5 e^-Å^{-2}$, which is believed to be the limiting

dose due to radiation damage for features near-atomic resolution[16,19,40,41]. The electron dose used in practice is typically a fold higher than the limiting dose. Although the additional dose does not contribute to the structure factor amplitudes at near-atomic resolution, it may have increased the signal up to the resolution limit of the final map, thus making the determination of particle parameters easier[16]. This conjecture agrees with the observation in the study of micrograph movie stack dose weighting, which found that only the initial few frames, not the subsequent frames, contribute to near-atomic features[42–44]. Finally, $\sigma_e = 0.004$ Å² is the elastic cross-section for carbon at 300 kV[45].

The overall temperature factor, or Rosenthal and Henderson's B-factor, is the dominant factor in estimating the theoretical limit. Here, we proposed a simplified assumption that limits only exist on instruments (TEM and electron detector) and that no other resolution-limiting factors exist. In other words, we assumed that all other procedures or techniques were ideal. For example, vitrified non-amorphous ice is perfectly flat and of ideal thickness, there is no beam-induced motion, and orientations of particles follow a uniform distribution, and there is no electron-charging effect. Therefore, B-factor represents a summary of all resolution-limiting factors of a given electron microscope and describes the overall quality of the instrumental setup. Holger Stark and his colleagues have summarized the current knowledge on existing state-of-the-art commercial EM hardware and their B-factors[46]. For the standard Titan Krios, they concluded that its B-factor is 50 Å², which was determined by re-evaluating data from EMPIAR-10216 as described by[47], with modifications to account for off-axial aberrations by splitting the micrographs into nine subsets[48]. Therefore, we computed the theoretical number of particle limits at $B = 50$ Å² (Table 2, column D). The sizes of the finest subsets obtained by CryoSieve were compared with such theoretical limits (Table 2, column E).

Out of the eight datasets examined, three (pfCRT, TSHR-Gs and apoferritin) were found to be close to their theoretical limits (Table 2, column E, emphasized by bold font). However, the TRPA1 dataset fell short of the theoretical limit by approximately 22-fold. This could be due to the lower resolution capabilities of the TF30 Polara TEM used in the study compared to more advanced models like the Titan Krios. It is possible that the assumed B-factor of 50 Å² for the TF30 Polara is relatively low and does not accurately reflect the properties of the TEM. Moreover, the sample preparation techniques employed during the TRPA1 study in 2015 might not have been fully optimized to attain the highest possible resolution. Hemagglutinin also fell short of the theoretical limit by roughly a factor of 36 due to using a tilt-collection strategy to compensate for the preferred orientation, which resulted in a larger effective ice thickness and a degradation in the quality of particle images. Lastly, LAT1 and TRPM8 exceeded the theoretical limit by factors of 9.8 and 6.3, respectively, suggesting that improvements in sample preparation could be made for these datasets.

## Discussion

In this study, we introduced the CryoSieve algorithm, which has the ability to estimate the minimum number of particles in a dataset, referred to as the finest subset. CryoSieve demonstrated that most

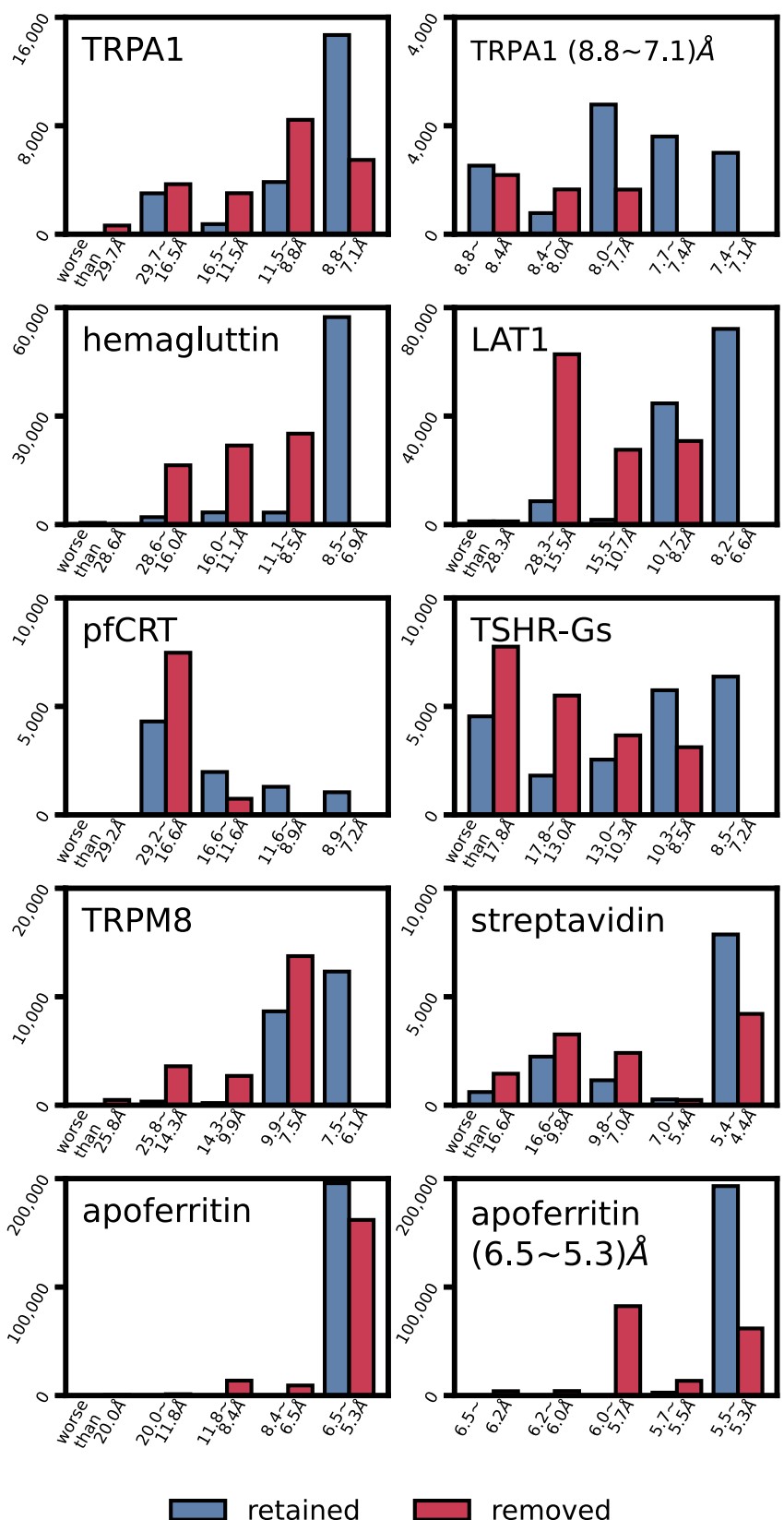

**Fig. 3 | The two-dimensional resolution distribution between retained and removed particles was compared.** The third iteration of CryoSieve achieved a retention ratio of 51.2% and a removal ratio of 48.8%, resulting in a similar number of particles for retention and removal. These two categories underwent CryoSPARC 2D clustering and averaging, i.e., 2D classification, with the number of 2D classes set to 50. All eight experimental datasets were tested. CryoSPARC reported the 2D resolution of each 2D class, along with the number of particle images belonging to it. We statistically analyzed the number of particles belonging to each 2D resolution and plotted histograms, demonstrating the difference between retained (steel blue) and removed (crimson) particles in terms of 2D resolution distribution. For TRPA1 and apoferritin, the bar with the highest resolution range was further finely divided and then plotted in a histogram, which is displayed to the right of the global histogram.

**Table 2 | The finest subsets alleviate high-resolution amplitude decay, along with a comparison to their theoretical number of particle limit**

| Dataset | A | B | C | D | E | F | G | H |
|---|---|---|---|---|---|---|---|---|
| TRPA1 | 3.90 | 141.9 | 78.1 (63.8–) | 198.5 | 147.3 (51.2–) | 521 | 11,426 (21.9×) | 43,585 (83.7×) |
| hemagglutinin | 3.62 | 232.0 | 160.8 (71.2–) | 226.9 | 146.4 (80.5–) | 975 | 34,078 (35.6×) | 130,000 (133.3×) |
| LAT1 | 3.11 | 132.6 | 96.0 (36.6–) | 147.3 | 94.9 (52.4–) | 6697 | 65,687 (9.8×) | 250,712 (37.4×) |
| pfCRT | 3.37 | 85.1 | **49.5 (35.6–)** | 235.8 | **66.8 (169.0–)** | 4212 | **4429 (1.01×)** | 16,905 (4.0×) |
| TSHR-Gs | 2.96 | 92.9 | **61.7 (31.2–)** | 96.9 | **62.4 (34.5–)** | 9205 | **13,465 (1.46×)** | 41,054 (4.5×) |
| TRPM8 | 2.98 | 94.7 | 76.7 (18.0–) | 110.1 | 82.2 (27.9–) | 2200 | 13,789 (6.3×) | 42,040 (19.1×) |
| apoferritin | 1.81 | 70.5 | **58.0 (12.5–)** | 81.6 | **49.2 (32.4–)** | 74,530 | **80194 (1.08×)** | 382,391 (5.1×) |
| streptavidin | 2.99 | 125.6 | 101.8 (23.8–) | 90.4 | 69.3 (21.1–) | 2152 | 7772 (3.6×) | 23,991 (11.1×) |

**A** Half-maps resolution of the CryoSieve-retained particles (Å); **B** B-factor reported by CryoSPARC auto-postprocessing obtained from all particles in the final stacks (Å²); **C** B-factor reported by CryoSPARC auto-postprocessing obtained from the CryoSieve-retained particles with temperature decrease (compared with all particles) in brackets (Å²); **D** Rosenthal's B-factor obtained from all particles in the final stacks (Å²); **E** Rosenthal's B-factor from the CryoSieve-retained particles with temperature decrease (compared with all particles) in brackets (Å²); **F** theoretical number of particles limit at $B = 50$ Å²; **G** number of the CryoSieve-retained particles with folds of theoretical limit in brackets.; **H** number of particles in final stacks with folds of theoretical limit in brackets. Three datasets (pfCRT, TSHR-Gs and apoferritin) were emphasized by bold font as the number of particle images in the finest subset approaches the theoretical limit.

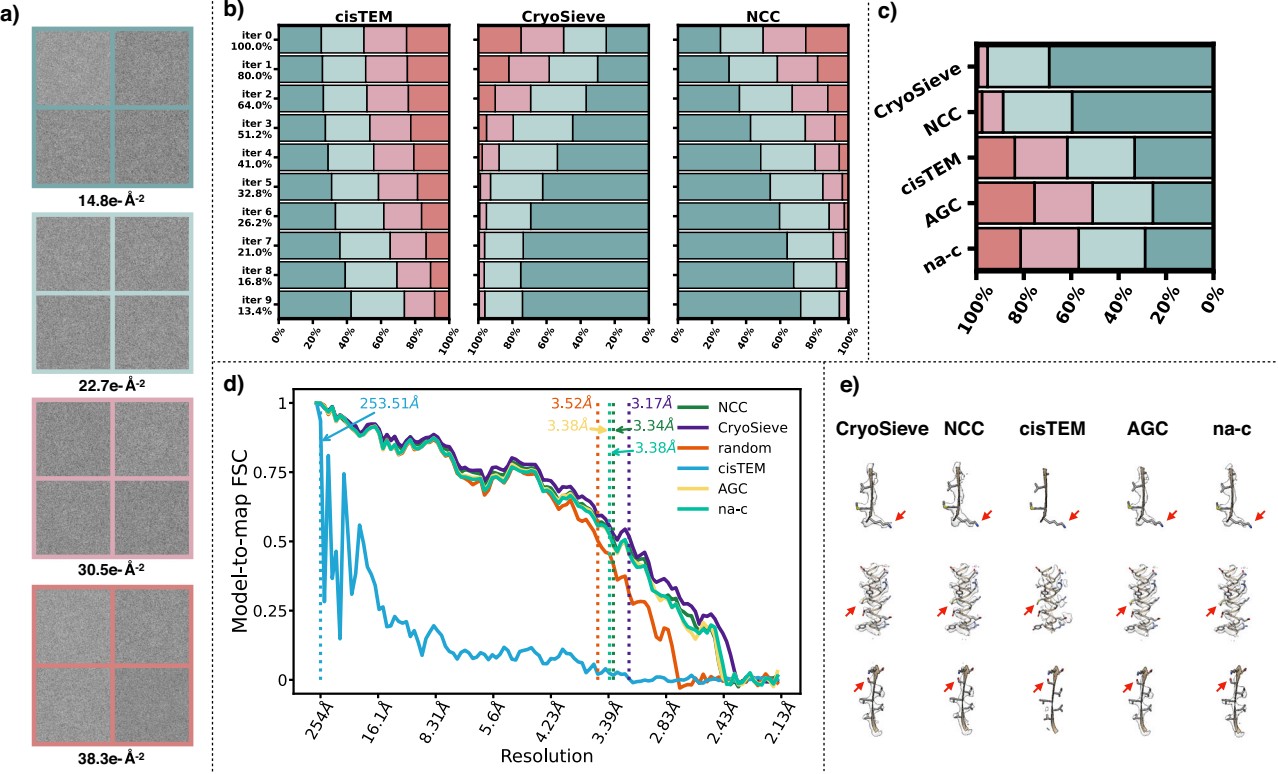

**Fig. 4 | CryoSieve prioritizes the removal of radiation-damaged particles.**
**a** Particles were selected from micrograph movie stacks of the proteasome, with each stack containing 32 frames and a total electron dose of 50 $e^-$Å$^{-2}$. This electron dose was uniformly distributed across all frames. Particles were extracted from consistent positions, using averages from frames 5–14, 10–19, 15–24, and 20–29. The average electron doses absorbed are denoted at the bottom, and four representative particles are displayed for each radiation damage level. **b** The graph depicts the proportions of particles with varying levels of radiation damage (differentiated by colors) in the retained particles across various retention ratios (indicated on the left). A comparison was made between the particles retained by cisTEM (left horizontal bars), CryoSieve (middle horizontal bars), and NCC (right horizontal bars). CryoSieve consistently sieved out particles in a sequence from high to low radiation damage, demonstrating superior performance over both cisTEM and NCC. **c** Particle distribution across the four radiation damage levels was analyzed using iteration 6 (featuring a retention ratio of 26.2%) from CryoSieve, NCC, and cisTEM. The analysis also incorporated particles retained by the AGC and non-alignment classification methods, with retention ratios auto-determined for these methods. **d, e** The side chains of the density maps reconstructed by CryoSPARC, using retained particles, were compared alongside the corresponding model-to-map FSCs. This comparison utilized a retention ratio of 26.2% (from iteration 6) for CryoSieve, NCC, cisTEM, and random methods. The retention ratios for AGC and non-alignment classification were auto-determined. The intersection between the FSC threshold ($FSC = 0.5$) and the FSC curve is represented as a vertical dashed line.

particles in the final stacks are superfluous and do not contribute to reconstructing density maps. On the other hand, the minority of particles that remain in the final stacks yields superior high-resolution amplitude. We also discovered that for some datasets, the size of the finest subset comes close to the theoretical limit. Therefore, CryoSieve can, to some degree, provide insight into a long-standing question in the cryo-EM field: How close can we approach the theoretical limit in practice?

CryoSieve can potentially establish a metric for the quantitative evaluation of various sample preparation techniques by measuring image quality based on the gaps between the theoretical limits and the size of the finest subsets. One of the possible future directions is to address the variables encountered during sample and grid preparation and establish cause-and-effect relationships. Resolving these issues, among others, cryo-EM could become a more versatile and influential technology in structural biology, potentially addressing research questions and aiding the growth of methodologies as the field advances[49].

## Methods

### Details of comparing the performance of particle sorting algorithms

Since cryo-EM single-particle image processing software has experienced rapid development in the past few years, some of the final stacks deposited in EMPIAR can be better processed by state-of-the-art algorithms. To eliminate effects from different refinement software and their versions, ensuring fair comparisons between various particle sorting algorithms, the final stacks deposited on EMPIAR were reprocessed under a standard workflow using CryoSPARC v4.1.0 following a standard workflow. For hemagglutinin, the initial model was generated by low-pass filtering its atomic model to 30 Å, while for the other proteins, initial models were generated by arbitrary random initialization using CryoSPARC. Then, uniform refinement was applied for TRPA1, TRPM8, hemagglutinin, LAT1, and apoferritin, while non-uniform refinement was applied for pfCRT and TSHR-Gs. For streptavidin, we employed local refinement. This was potentially due to the use of a phase plate in the streptavidin dataset, as ab initio reconstruction failed to produce a density map for streptavidin.

To enable unbiased comparisons of density maps before and after particle sorting, the retained particles obtained from each particle sorting algorithm underwent identical refinement procedures, as previously described using CryoSPARC v4.1.0 in the standard workflow. The reconstructed density maps were used for subsequent measurements. To ensure that there's no undue influence of information from the discarded particles via their contribution to pose estimation, the former Euler angles were discarded (except streptavidin), and new sets of Euler angles were determined through the refinement of the retained particles. Moreover, in order to maintain independence between the two half sets and ensure that the Fourier Shell Correlation (FSC) served as the golden standard, half-set splits were preserved throughout the subsequent procedure by turning off the option "Force re-do GS split".

The reconstructed density maps were evaluated by several metrics, including FSC-based resolution, Q-score and Rosenthal-Henderson B-factor. CryoSPARC produced two raw half maps and an auto-postprocessed density map (FSC-weighted, B-factor sharpened, two half sets averaged), accompanied by reporting half-maps FSC.

FSC-based metric includes half-maps FSC (directly reported by CryoSPARC) and model-to-map FSC. Map-to-model FSC resolution was calculated using the following procedure, with the auto-postprocessed density map as input. The corresponding atomic model of the dataset was converted to the ground-truth density map by the molmap function of Chimera at Nyquist resolution. The mask was generated from the ground-truth density map (after low-pass filtering to 8 Å, extending by 4 pixels and applying a cosine-edge of 4 pixels) using RELION. Model-to-map FSC curves were determined between the input density map (obscured by the mask) and the ground-truth density map. The resolution threshold of the map-to-model FSC was set to 0.5.

As Q-score is sensitive to B-factor sharpening, the Q-scores of both the raw maps and the auto-postprocessed maps were measured. The auto-postprocessed maps were directly provided by CryoSPARC, while the raw maps were obtained by first averaging the two raw half maps provided by CryoSPARC, then low-pass filtering them to an appropriate resolution, in order to eliminate the impact of varying noise intensities on the density maps. The low-pass filtering threshold frequency ranged from 0.3 Å to 0.5 Å higher than the CryoSPARC reported half-maps FSC resolution, thus ensuring the retention of useful signals. Specifically, the threshold frequency for TRPA1 was 3.5 Å, for TRPM8 and TSHR-Gs it was 2.7 Å, for hemagglutinin it was 3.4 Å, for pfCRT it was 3.0 Å, for apoferritin it was 1.6 Å, for streptavidin it was 2.8 Å, and for LAT it was 2.8 Å. Q-score was calculated using the MAPQ plugin for UCSF Chimera, with all parameters set to their default values.

Rosenthal-Henderson B-factors were determined by fitting the formula that describes the relationship between resolution and the number of particles used for reconstruction. Five half-splitting repetitions were adopted for each dataset. After each repetition, the Euler angles were re-estimated by CryoSPARC, and the reported resolution was used for data fitting.

All conversions between CryoSPARC and RELION were performed using the pyem script.

### CryoSieve's parameters

CryoSieve iteratively performs 3D reconstruction and particle sieving, while maintaining independence between two half sets by independently sieving each set of particles. 3D reconstructions of each subset were performed using RELION v4.0-beta-2, with the option "−−subset" to preserve the half-set splitting. A mask, generated from the atomic model using RELION (low-pass filtered to 8 Å), was applied to the reconstructed raw density map to obtain $x^{(k-1)}$ in Eq. 2 of the CryoSieve score. The same mask was applied to other particle sorting algorithms such as NCC and AGC, to ensure fair comparisons. Subsequently, particles were sieved out based on the ascending order of the CryoSieve score. In total, nine iterations were carried out, with each iteration retaining 80% of the particles from the previous iteration. The cutoff frequency of the highpass operator $H^{(k)}$ increased linearly as the iteration progressed. For all datasets, except for LAT1 and apoferritin, the initial cutoff frequency was set at 40 Å, and the final cutoff frequency was 3 Å. For LAT1, the initial cutoff frequency was 50 Å, and the final cutoff frequency was also 3 Å. For apoferritin, the initial cutoff frequency was 40 Å, and the final cutoff frequency was also 2 Å.

### Reporting summary

Further information on research design is available in the Nature Portfolio Reporting Summary linked to this article.

## Data availability

The raw final stack datasets analyzed in this study were downloaded from the EMPIAR repository using accession codes EMPIAR-10024, EMPIAR-11233, EMPIAR-10097, EMPIAR-11120, EMPIAR-10264, EMPIAR-10330, EMPIAR-10269, EMPIAR-10200. Atomic coordinates from Protein Data Bank 6PCQ were used for the generation of simulated particles using InSilicoTEM v2.1.0. Source data are provided with this paper.

## Code availability

CryoSieve[22] is now open-sourced and available on GitHub [https://github.com/mxhulab/cryosieve]. A detailed tutorial can also be found on its homepage. Moreover, datasets used in this manuscript, along with the expected outputs after running CryoSieve, have been deposited on GitHub and can be accessed via CryoSieve's homepage. Code has been uploaded to Zenodo and can be accessed via [https://doi.org/10.5281/zenodo.10040463].

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

## Acknowledgements

This work was supported by the National Key R&D Program of China (No. 2021YFA1001300) (to C.B.), the National Natural Science Foundation of China (No. 12271291) (to C.B.), the Advanced Innovation Center for Structural Biology (to M.H.), the Beijing Frontier Research Center for Biological Structure (to M.H.), Shenzhen Academy of Research and Translation (to M.H.), Natural Science Foundation of China (No. 12071244) (to Z.S.). We would like to express our gratitude to Shouqing Li and Ranhao Zhang for generously sharing their expertise in particle selection and density map reconstruction in Cryo-EM. Our thanks also go to Dr. Nan Liu for providing valuable suggestions on this work, and to Jie Xu for his assistance in constructing the real radiation damage dataset of the proteasome.

## Author contributions

C.B., M.H., and Z.S. initiated the project. M.H., Q.Z., and J.Z. developed CryoSieve and carried out testing. H.Z. provided support in using InSilicoTEM. J.Z. and M.H. analyzed the data. M.H., J.Z., and C.B. wrote the manuscript.

## Competing interests

The authors declare no competing interests.
