## [Peer Review File · Nature Communications]

A minority of final stacks yields superior amplitude in single-particle cryo-EMReviewers' Comments:

Reviewer #1:

Remarks to the Author:

The manuscript "Not final yet: a minority of final stacks yields superior amplitude in single-particle cryo-EM" from Zhu et al approaches the important question in cryo-EM of the number of images required to get high resolution structures for a protein of a given size, for which practical experience and theoretical limits seem at odds with each other. However, the paper in it's current state could be improved to make the method clearer and the impact of the results more significant.

First of all, from the initial description it is not abundantly clear what the CryoSieve algorithm is. First it seems to be reconstruction algorithm independent, as it indicates that it uses 'a cryo-EM single-particle reconstruction software selected by the user'. While this is nice, it begs the reader to ask if different reconstruction software produce different results with this algorithm. Furthermore it leads one to wonder if the iterations must be done by hand or if CryoSieve has been setup to work automatically with one software or another (cryoSPARC, Relion, or cisTEM perhaps). The existence of such a pipeline would be made somewhat clearer with the sharing of the software itself which it has been declared "will be open-source upon publication and is also available upon request during the review process".

The second aspect of the algorithm which is very unclear is despite a mathematical formalism presented for the cryoSieve score, it is very unclear how various important parameters within this score are chosen per dataset and per iteration within each dataset. Two clear examples are 1) the choice of high pass filter per iteration and 2) the score for which particles are chosen to be retained or dropped. These seem to vary as a function of dataset, as described in the methods, and it is unclear if this has been done in a systematic way. These choices also make it very unclear how this method can be compared to the other methods (random, NCC, AGC, and cisTEM). I think the algorithm would be much clearer if these parameter choices per iteration were explicitly described within the mathematical formalism presented, and if the other methods (especially the 'random' method to which a clear comparison can be made) can also be described within this same formalism so that it is much clearer what is actually being done. In this light, the cisTEM method has been clearly separated from the NCC and AGC methods, and I think keeping them together would improve the correny of the arguments of this paper.

Another aspect of this algorithm that is not clear is how it relates to sieve-based strategies in machine learning, or if indeed has no relation to these, and the name was not chosen in relation to other sieve-based algorithms. For example see "Universal sieve-based strategies for efficient estimation using machine learning tools" arXiv:2003.01856v2 from Qiu, Luedtke, and Carone (2020).

Despite the lack of clarity in the description of the algorithm - the results of the CryoSieve method presented in this paper are enlightening - that similar resolution data can be achieved with 26.2%-32.8% of the data is quite interesting. But what the consequences of this result are could be discussed in much greater detail. The theoretical experiments show that CryoSieve is dropping particles with increased radiation damage (according to one model of simulated radiation damage), which is interesting - but is this the only aspect of the images the method is picking up on? Because the CryoSieve score depends so critically on the high pass filter - are there lessons to be learned regarding how to pick particles or collect data so that we can actually increase our resolution with the same number of particles in the future? Or even guidance for experimentalists for how to collect fewer of these 'futile particles'?

It is impressive to see the comparisons of the CryoSieve results to the theoretical limits, however the language in the text needs clarification 'the TRPA1 dataset fell short by approximately 52' should be '52-fold' I believe. Otherwise this is quite misleading. That the pFCRT dataset matches the theoretical limit very closely is very exciting and it would be worthwhile to hear more discussion about why the

authors think this dataset outperforms the results of the others and how one might achieve similarly good results for the other datasets either through further optimization of cryoSieve or changes in the data collection procedure (or some third approach).

Overall, while this manuscript shows valuable results within the context of understanding which particles aren't contributing information to high resolution reconstructions of molecules in single particle cryoEM, I believe it would need some significant clarifications and improvements before it can be accepted for publication by this journal.

Minor points:

It is advisable to cite EMPIAR: Iudin A, Korir PK, Somasundharam S, Weyand S, Cattavittello C, Fonseca N, Salih O, Kleywegt GJ, Patwardhan A (2023). "EMPIAR: the Electron Microscopy Public Image Archive." *Nucleic Acids Res.*, 51, D1503-D1511. <https://doi.org/10.1093/nar/gkac1062>.

In section 2.1 the sentence "We have demonstrated that the CryoSieve score can identify particles with incorrect pose parameters or components in the high-frequency range through theoretical analysis and simulation verification." should include references to the sections which do these theoretical analyses and simulation verifications.

In Figure 1 - it seems like a substantial coincidence that four datasets use 26.2% of the data while two datasets use 32.8% of the data to achieve the final high resolution results while culling particles with the CryoSieve method - is this an artifact of the way iterations are chosen or a typo? If it is an artifact of the method it would be worthwhile to clarify this with the more in depth description of the method requested above.

Reviewer #2:

Remarks to the Author:

The manuscript by Zhu, et al. describes their work using high-frequency signals to sort particles for finding a minimal "finest subset" for 3D reconstruction in cryo-EM. The work aims to address a crucial question in cryo-EM: that is, how to get the minimum number of particles required to reach a specific resolution. To do so, they developed a CryoSieve procedure and applied it to six EMPIAR data sets with resolutions ranging from 4.11 to 3.04 Å. They show that with the CryoSieve procedure, they can identify and remove the majority of particles while maintaining a sufficient number of particles to reach a similar or higher resolution as evaluated by FSC curves and Q scores. The work is interesting in the sense that it provides a practical way to sort particles. With the six data sets selected, their technique is sound, and the results support their main claim. However, as a new method, it must be validated using a wide range of data sets to show its broad applicability. This is particularly true because many data sets can be downloaded from EMPIAR. Below are my comments for the authors to consider.

Major:

1. The authors developed a CryoSieve procedure for particle sorting to further remove particles after consensus refinement. They selected six EMPIAR data sets with resolutions between 4.11 and 3.04 Å. To show its broader applicability, the authors should expand their tests on additional data sets. 1) They should include data sets better than 3 Å, for example selecting data sets between 2-3 Å, and data sets better than 2.0 Å. As the resolution of cryo-EM has reached atomic resolutions at about 1.2 Å, it would be interesting to see if CryoSieve works on these very high-quality datasets. The analysis will help better understand the performance of CryoSieve. 2) The signal over noise of particles depends on the particle size. To evaluate the quality of the work and performance of CryoSieve, it is suggested to evaluate the performance of CryoSieve in dealing with particles of different sizes, i.e. molecular weights as kDa. Cryo-EM can allow structure determination of particles with a molecular

weight of about 50 kDa or lower. The authors should evaluate the performance of CryoSieve on such small particles and add molecular weight to Tables 1 or 2.

2. One challenge in cryo-EM is data heterogeneity. In solution, macromolecules are in equilibrium in many conformational states which are captured during the vitrification process. Cryo-EM data analysis is essentially a triage process to filter out conformational states and radiation damage. The author claims that CryoSieve can remove radiation-damaged particles. How can they exclude the possibility that the particles they removed could be particles belonging to minor conformational states which are slightly different from the consensus model? The authors used simulated particles to show the effectiveness of CryoSieve in removing radiation-damaged particles. They need to demonstrate the effectiveness using experimental data.

Minor:

1. P. 3, line 89: The authors used "high-resolution amplitude" for sorting particles in Fourier space. Have the authors sorted particles based on the high-resolution phase? It would be interesting to compare phase-based sorting with amplitude-based sorting.

2. Page 6, lines 151-156. The authors describe the 2D and 3D classification work used in reference 26. Such a statement does not bring in new information here and should be deleted.

3. Page 6, lines 164-166. The authors claim that CryoSieve can remove over half of the particles with unreliable high-frequency signals without negatively affecting the final reconstruction. However, it's not clear what's the criterion/threshold to define "unreliable high-frequency signals". Besides, after removing the "unreliable high-frequency signals", have the authors observed improved cryo-EM densities or structural features that were blurred or missing in the published maps?

4. Page 6, lines 168-171. The authors compared CryoSieve with two other sorting methods of NCC and AGC. The authors should discuss in more detail why their method is better than the other two. Did the authors observe the preferred orientation issue while sorting particles based on high-frequency signals? In cryo-EM, nonalignment classification is routine and effective for the classification of heterogeneous data. The nonalignment classification can sort and remove particles, in the meanwhile can identify additional conformational states. The authors should compare the performance of CryoSieve with the nonalignment classification in terms of removing particles while maintaining the resolution.

5. Page 6, lines 181-183. The authors used "Einstein-from-noise" to justify the removal of the deposited Euler angles. This statement is not accurate because there is no evidence that the published reconstructions/Euler angles suffered from the "Einstein-from-noise" issue. The authors should revise the sentence to say "to remove bias in the published maps".

6. Page 6, lines 185-187. The authors should include all metrics, in addition to FSC and Q-score, that they have used to evaluate the maps before and after the CryoSieve procedure. As B factors are important for evaluating data quality, they should plot B factors with respect to the number of interactions in Figure 2 and Supplementary Figure 2.

7. Page 7, Figure 1 caption: Did the authors apply the same B-factor to sharpen the maps before and after CryoSieve? In addition to the sharpened maps, the authors may compare the non-sharpened maps in a Supplementary Figure. For a better comparison, they should include the contour levels that were used to draw maps before and after CryoSieve.

8. Page 7, lines 194-195. If cisTEM reports a per-particle score, the authors should explain why the score can't be used as a particle sorting criterion.

9. Page 8, lines 211-214. Why did CryoSieve remove a substantial number of high-resolution 2D

particles in TRPA1, but not in the other five data sets (Figure 2)? The authors should perform 2D class averaging on additional data sets (see major #1).

10. Page 8, section 2.4: Using simulated data, the authors claim that CryoSieve can effectively detect radiation-damaged particles better than NCC and cisTEM. The authors should also compare CryoSieve performance with the AGC method and nonalignment classification method. In addition, they should use experimental data, not just simulated data to show its effectiveness in the treatment of experimental radiation damage.

11. Page 9, Table 2: For B-factor calculation, the authors should use the Rosenthal and Henderson's B-factor method instead of values from the cryoSPARC auto-processing.

12. Page 11, lines 330-345. The discussion on sample preparation is off-topic to the work and should be removed or revised in the context of CryoSieve.

13. The authors should have shared their code as an attachment for a better evaluation of the work.

Authors' Response to Reviews of

Not final yet: a minority of final stacks yields superior amplitude in single-particle cryo-EM

Jianying Zhu, Qi Zhang, Hui Zhang, Zuoqiang Shi, Mingxu Hu and Chenglong Bao
Submitted to *Nature Communications*, NCOMMS-23-22170-T

RC: *Reviewers' Comment*, AR: Authors' Response, □ Manuscript Text

We sincerely thank the valuable suggestions and comments from the reviewers. We list our point-to-point replies in the following context and hope that the revision can address the concerns.

Response to Referee #1

RC: *The manuscript "Not final yet: a minority of final stacks yields superior amplitude in single-particle cryo-EM" from Zhu et al approaches the important question in cryo-EM of the number of images required to get high resolution structures for a protein of a given size, for which practical experience and theoretical limits seem at odds with each other. However, the paper in it's current state could be improved to make the method clearer and the impact of the results more significant.*

AR: Thanks for your support of this paper. We have made substantial changes in the revised version and hope that it can clearly show the significance of CryoSieve.

RC: *(1-1) First of all, from the initial description it is not abundantly clear what the CryoSieve algorithm is. First it seems to be reconstruction algorithm independent, as it indicates that it uses 'a cryo-EM single-particle reconstruction software selected by the user'. While this is nice, it begs the reader to ask if different reconstruction software produce different results with this algorithm.*

AR: Thank you for your insightful comments and constructive suggestions. We have added a Supplementary Figure to clarify the performance comparison between using Relion and CryoSPARC in the reconstruction algorithm/module.

Our statement in the initial submission was based on our observation that the reconstructed density outputs from mainstream software were nearly indistinguishable, as evidenced by the high correlations among them (refer to Supplementary Fig. 2b). However, upon further investigation and in light of your valuable feedback, we were surprised to discover that using Relion in combination with CryoSieve yielded significantly superior results compared to CryoSPARC (see Supplementary Fig. 2a). We have accordingly updated our manuscript to reflect this new insight.

The observed performance difference when switching from Relion to CryoSPARC can be clarified as follows: It is a common observation among cryo-EM image processing researchers that the range of reconstructed maps from CryoSPARC differs from those from RELION. Specifically, CryoSPARC applies a multiplication factor to the amplitude of the reconstructed density (see Supplementary Fig. 2c). This action changes the scale between the reconstructed density map and the corresponding particles. Such a modification introduces a scale variation, which significantly impacts the computed CryoSieve score, rendering it less effective. The score is given by:

$$g_i = |Hb_i|_2^2 - |H(b_i - \tilde{A}_i x)|_2^2.$$

In essence, replacing x with $x' = \alpha X$, where $\alpha \neq 1$ (notably, in CryoSPARC, α is much greater than 1), disrupts the score. Also, estimating the scalar α in CryoSPARC is difficult due to the inaccessibility of the source code.

We have revised our manuscript as

Given that \$g_j\$ relies on the accurate amplitude of the reconstructed density map \$x^{(k)}\$, CryoSPARC is not the optimal choice for reconstruction (Supplementary Figure 2).

RC: *(1-2) Furthermore it leads one to wonder if the iterations must be done by hand or if CryoSieve has been setup to work automatically with one software or another (CryoSPARC, Relion, or cisTEM perhaps). The existence of such a pipeline would be made somewhat clearer with the sharing of the software itself which it has been declared "will be open-source upon publication and is also available upon request during the review process".*

AR: Thank you for your comment. We would like to clarify that users are not required to perform iterations manually. CryoSieve automates this process, with each iteration encompassing both reconstruction and sieving. The software takes the path of the reconstruction module from RELION as an input option and utilizes it in the iterative sieving process. This procedure is illustrated in a flow chart, available in Supplementary Figure 3. Moreover, CryoSieve is now open-sourced and available on GitHub <https://github.com/mxhulab/cryosieve>. A comprehensive user guide can be found on the project's homepage, providing detailed instructions and assistance for users. We have revised our manuscript as

A flow chart scheme is provided in Supplementary Figure 3.

and

CryoSieve is now open-sourced and available on GitHub (<https://github.com/mxhulab/cryosieve>). A detailed tutorial can also be found on its homepage. Moreover, datasets used in this manuscript, along with the expected outputs after running CryoSieve, have been deposited on GitHub and can be accessed via CryoSieve's homepage.

RC: *(2) The second aspect of the algorithm which is very unclear is despite a mathematical formalism presented for the cryoSieve score, it is very unclear how various important parameters within this score are chosen per dataset and per iteration within each dataset. Two clear examples are 1) the choice of high pass filter per iteration and 2) the score for which particles are chosen to be retained or dropped. These seem to vary as a function of dataset, as described in the methods, and it is unclear if this has been done in a systematic way. These choices also make it very unclear how this method can be compared to the other methods (random, NCC, AGC, and cisTEM). I think the algorithm would be much clearer if these parameter choices per iteration were explicitly described within the mathematical formalism presented, and if the other methods (especially the 'random' method to which a clear comparison can be made) can also be described within this same formalism so that it is much clearer what is actually being done.*

AR: Thanks for your comment. We have given the parameter settings for CryoSieve and the other comparative algorithms in Supplementary Material VI. For CryoSieve, the high-pass cutoff frequency increases linearly across iterations. Additionally, for CryoSieve, NCC, cisTEM, and random, a fixed retention ratio of 80% was maintained in our experiments.

We have revised our manuscript as

The high-pass cutoff frequency of CryoSieve increases linearly across iterations.

The parameter settings for CryoSieve and the other comparative algorithms were listed in Supplementary Material VI.

along with adding Supplementary Material VI.

RC: *(3) In this light, the cisTEM method has been clearly separated from the NCC and AGC methods, and I think keeping them together would improve the correny of the arguments of this paper.*

AR: Thanks for your suggestion. We have integrated the cisTEM results from the supplementary section and combined them with results from other algorithms, namely CryoSieve, NCC, AGC, and non-alignment classification, using random as the baseline for comparison. It is important to emphasize that while cisTEM can report a score for each individual particle image, this score is provided after its 3D refinement. The pose parameters of the particles undergo re-estimation or refinement during cisTEM's 3D refinement process. Given the differences in alignment and other image processing workflows between cisTEM and CryoSPARC, a direct comparison between cisTEM and CryoSieve may not be fair.

RC: *(4) Another aspect of this algorithm that is not clear is how it relates to sieve-based strategies in machine learning, or if indeed has no relation to these, and the name was not chosen in relation to other sieve-based algorithms. For example see "Universal sieve-based strategies for efficient estimation using machine learning tools" arXiv:2003.01856v2 from Qiu, Luedtke, and Carone (2020).*

AR: Thank you for your comment. CryoSieve is a software designed to filter out non-essential particles by leveraging high-frequency distance. However, its method does not directly relate to the sieve-based estimations discussed in the suggested paper. In that context, a sieve estimator employs a sequence of simpler models (the sieves) to approximate a complex model. While both utilize the concept of a 'sieve,' the application and methodology differ significantly.

RC: *(5-1) Despite the lack of clarity in the description of the algorithm - the results of the CryoSieve method presented in this paper are enlightening - that similar resolution data can be achieved with 26.2%-32.8% of the data is quite interesting.*

AR: Thank you for your support.

RC: *(5-2) But what the consequences of this result are could be discussed in much greater detail. The theoretical experiments show that CryoSieve is dropping particles with increased radiation damage (according to one model of simulated radiation damage), which is interesting - but is this the only aspect of the images the method is picking up on?*

AR: Thank you for your suggestion. We have simulated orientation, translation and CTF parameter errors in the TRMP8 dataset, and found out that CryoSieve is capable of efficiently removing particles achieving a high accuracy of over 90%. We organized the result as Supplementary Material III. However, non-alignment classification seems to achieve comparable accuracy in cases of the simulated orientation, translation and CTF parameter error (also in Supplementary Material III). Therefore, these type of errors are unlikely to present in the final stacks.

Generally, we cannot definitively determine the full range of image features that CryoSieve identifies. Current experimental evidence indicates that CryoSieve preferentially eliminates particles experiencing radiation damage. However, it's unlikely that radiation-damaged particles account for all the eliminations. This question might be better addressed by analyzing their coordinates in the micrographs considering not only the X-Y

axis but the Z-axis as well. Factors such as air-water interference or charging effects could play roles. This is an intriguing area that warrants further investigation.

We have revised our manuscript as

It's worth noting that CryoSieve can efficiently remove particles with incorrect pose and CTF parameter estimations, achieving a high accuracy of over 90% (Supplementary Material III). However, These particles are also removed by the non-alignment classification approach (Supplementary Material III), making them unlikely to be present in the final stacks.

and add Material III in Supplementary.

RC: *(5-3) Because the CryoSieve score depends so critically on the high pass filter - are there lessons to be learned regarding how to pick particles or collect data so that we can actually increase our resolution with the same number of particles in the future? Or even guidance for experimenters for how to collect fewer of these 'futile particles'?*

AR: Current particle-picking strategies, encompassing both template-based and deep-learning-based methods, predominantly rely on the low-frequency information of the target biological macromolecule. Given that the CryoSieve score is contingent on the high-pass filter, this highlights the value of maintaining a high retention level in the high-frequency range when evaluating particle quality. While attempts aim to assess particle quality during the data acquisition phase, minimizing the collection of these "futile particles" remains a persistent, unresolved challenge in the cryo-EM field, necessitating further research.

An additional insight derived from CryoSieve suggests substantial potential for improving sample preparation techniques to reduce the proportion of futile particles. Following the submission of this paper, we collaborated with Professor Hongwei Wang, a renowned expert in sample separation. Our collaboration revealed that the conditions during sample preparation play a pivotal role in limiting the presence of futile particles.

RC: *(6) It is impressive to see the comparisons of the CryoSieve results to the theoretical limits, however the language in the text needs clarification 'the TRPA1 dataset fell short by approximately 52' should be '52-fold' I believe. Otherwise this is quite misleading.*

AR: Thank you for your support and pointing that out. We have revised it.

RC: *(7) That the pfCRT dataset matches the theoretical limit very closely is very exciting and it would be worthwhile to hear more discussion about why the authors think this dataset outperforms the results of the others and how one might achieve similarly good results for the other datasets either through further optimization of cryoSieve or changes in the data collection procedure (or some third approach).*

AR: Thank you for your feedback. It is excited to see that pfCRT is nearing the theoretical limit. Notably, during its data collection, a Cs corrector and energy filter were used in tandem. Only a few facilities possess both these devices, and it's possible that the pfCRT dataset benefited from this combination. However, whether the advancements in the Cs corrector, energy filter, and sample preparation contribute to obtaining datasets closer to the theoretical limit and to what extent they help still largely requires further research.

Given that this matter requires further investigation and is beyond the scope of this work, we choose to avoid making strong claims about how to reach the theoretical limit in this manuscript.

RC: *(8) Overall, while this manuscript shows valuable results within the context of understanding which particles aren't contributing information to high resolution reconstructions of molecules in single particle cryoEM, I believe it would need some significant clarifications and improvements before it can be accepted*

for publication by this journal.

AR: Thank you for your valuable comments. We hope the revised manuscript can address your concerns.

RC: (9) *It is advisable to cite EMPIAR: Iudin A, Korir PK, Somasundharam S, Weyand S, Cattavittello C, Fonseca N, Salih O, Kleywegt GJ, Patwardhan A (2023). "EMPIAR: the Electron Microscopy Public Image Archive." Nucleic Acids Res., 51, D1503-D1511. <https://doi.org/10.1093/nar/gkac1062>.*

AR: Thank you for your suggestion. We added this citation.

RC: (10) *In section 2.1 the sentence "We have demonstrated that the CryoSieve score can identify particles with incorrect pose parameters or components in the high-frequency range through theoretical analysis and simulation verification." should include references to the sections which do these theoretical analyses and simulation verifications.*

AR: We have added the reference section in the revision. In Supplementary Material I and III, we carry out two types of analysis:

- Assuming that noise in particles follows a Gaussian distribution, we have shown that, with high probability, the CryoSieve score is an ideal indicator of particle image quality, distinguishing it from typical cryo-EM damage or artifact, including high-frequency random phasing and inaccurate estimation of imaging parameters such as rotation angle, in-plane translation, and CTF parameters.
- When we use simulated datasets for particle sieving, CryoSieve score exhibits remarkable accuracy in removing particles with incorrect pose and CTF parameter estimations, achieving a precision rate of over 90%.

We have revised our manuscript as

~~We have demonstrated that the CryoSieve score can identify particles with incorrect pose parameters or components in the high-frequency range through theoretical analysis and simulation verification. Assuming that noise in particles follows a Gaussian distribution, we have shown that, with high probability, the CryoSieve score is an ideal indicator of particle image quality, distinguishing it from typical cryo-EM damage or artifacts (Supplementary Material I). Furthermore, when simulating radiation damage as high-frequency random phasing, the CryoSieve score exhibits remarkable accuracy in selecting particles even with a very low signal-to-noise ratio (approximately 0.001), achieving a precision rate of around 90% (Supplementary Material I).~~

We have demonstrated that the CryoSieve score can identify particles with incorrect pose parameters or components in the high-frequency range through theoretical analysis and simulation verification (Supplementary Material I and III). Specifically, assuming that noise in particles follows a Gaussian distribution, we have shown that, with high probability, the CryoSieve score is an ideal indicator of particle image quality, distinguishing it from typical cryo-EM damage or artifacts (Supplementary Material I). Furthermore, the CryoSieve score exhibits remarkable accuracy in removing particles with incorrect pose and CTF parameter estimations, achieving a high accuracy of over 90% (Supplementary Material III).

RC: (11) *In Figure 1 - it seems like a substantial coincidence that four datasets use 26.2% of the data while two datasets use 32.8% of the data to achieve the final high resolution results while culling particles with the CryoSieve method - is this an artifact of the way iterations are chosen or a typo? If it is an artifact of the*

method it would be worthwhile to clarify this with the more in depth description of the method requested above.

AR: In all experiments, we have set the retention ratio for each iteration, a hyperparameter of CryoSieve, to 80%. If the finest subset is in iteration 5, its ratio would be $0.8^5 = 32.8\%$. If the finest subset is in iteration 6, the ratio becomes $0.8^6 = 26.2\%$. The iteration in which the finest subset appears is determined based on comprehensive metrics, including FSC, Q-score and Rosenthal-Henderson B-factor, as illustrated in Figure 2.

Response to Referee #2

RC: *The manuscript by Zhu, et al. describes their work using high-frequency signals to sort particles for finding a minimal “finest subset” for 3D reconstruction in cryo-EM. The work aims to address a crucial question in cryo-EM: that is, how to get the minimum number of particles required to reach a specific resolution. To do so, they developed a CryoSieve procedure and applied it to six EMPIAR data sets with resolutions ranging from 4.11 to 3.04 Å. They show that with the CryoSieve procedure, they can identify and remove the majority of particles while maintaining a sufficient number of particles to reach a similar or higher resolution as evaluated by FSC curves and Q scores. The work is interesting in the sense that it provides a practical way to sort particles. With the six data sets selected, their technique is sound, and the results support their main claim. However, as a new method, it must be validated using a wide range of data sets to show its broad applicability. This is particularly true because many data sets can be downloaded from EMPIAR. Below are my comments for the authors to consider.*

AR: Thanks for your support. We appreciate your suggestions for improving our paper.

RC: *(1-1) The authors developed a CryoSieve procedure for particle sorting to further remove particles after consensus refinement. They selected six EMPIAR data sets with resolutions between 4.11Å and 3.04Å. To show its broader applicability, the authors should expand their tests on additional data sets. 1) They should include data sets better than 3 Å, for example selecting data sets between 2-3 Å, and data sets better than 2.0 Å. As the resolution of cryo-EM has reached atomic resolutions at about 1.2 Å, it would be interesting to see if CryoSieve works on these very high-quality datasets. The analysis will help better understand the performance of CryoSieve.*

AR: Thank you for the thoughtful suggestion. We introduced an additional dataset of human apoferritin (EMPAIR-10200) at 1.9Å for further investigation. Impressively, CryoSieve was able to filter out 79% of the particles from the final stack, enhancing the resolution from 1.89Å to 1.81Å (based on half-maps resolution). Additionally, the number of particles in the finest subsets exceeded the theoretical limit by only a slight margin of 8%.

To achieve atomic resolutions around 1.2Å, it becomes necessary to consider high-order aberrations during image processing and density map reconstruction. However, the current version of CryoSieve does not implement high-order aberrations correction. The potential for CryoSieve to operate at these cutting-edge resolutions is intriguing, and we plan to explore this in future research.

The seventh is from human apoferritin (EMPIAR-10200),...

Out of the six datasets examined, two (pfCRT and TSHR-Gs) were found to be close to their theoretical limits (Table 2, column E, emphasized by bold font).

Out of the eight datasets examined, three (pfCRT, TSHR-Gs and apoferritin) were found to be close to their theoretical limits (Table 2, column E, emphasized by bold font).

RC: *(1-2) The signal over noise of particles depends on the particle size. To evaluate the quality of the work and performance of CryoSieve, it is suggested to evaluate the performance of CryoSieve in dealing with particles of different sizes, i.e. molecular weights as kDa. Cryo-EM can allow structure determination of particles with a molecular weight of about 50 kDa or lower.*

AR: Thank you for your valuable suggestion. We have included the streptavidin dataset (EMPAIR-10269) in Tables 1, 2 and Figures 1, 2, 3, which has a molecular weight of 52kDa. CryoSieve managed to remove 67.2%

of the particles from the final stack, improving the resolution from 3.15Å to 2.99Å (based on half-maps resolution).

RC: *(1-3) The authors should evaluate the performance of CryoSieve on such small particles and add molecular weight to Tables 1 or 2.*

AR: Thank you for your suggestion. We've incorporated a streptavidin dataset (52kDa) and applied CryoSieve for particle sieving. The results have been added to Tables 1 and 2. Additionally, we've included the molecular weight for each entry in Table 1.

RC: *(2) One challenge in cryo-EM is data heterogeneity. In solution, macromolecules are equilibrium in many conformational states which are captured during the vitrification process. Cryo-EM data analysis is essentially a triage process to filter out conformational states and radiation damage. The author claims that CryoSieve can remove radiation-damaged particles. How can they exclude the possibility that the particles they removed could be particles belonging to minor conformational states which are slightly different from the consensus model?*

AR: Thank you for your valuable comments. The main goal of our manuscript is to develop a numerical method capable of identifying the smallest subset within the final stack without losing the resolution, which does not specifically address the challenges associated with heterogeneity. Our experiments demonstrate that if particles contain radiation damage or parameter estimation errors (such as orientation, translation, or CTF), CryoSieve can accurately and robustly identify them. Nonetheless, in practical applications, we cannot guarantee that the particles discarded do not contain information about other conformations. We can only ensure that they are unnecessary for the reconstructed density map. In the future, it is promising to delve into the heterogeneity problem by integrating CryoSieve with classification techniques.

RC: *(3) The authors used simulated particles to show the effectiveness of CryoSieve in removing radiation-damaged particles. They need to demonstrate the effectiveness using experimental data.*

AR: Thank you for your suggestion. In the revision, we have transitioned from utilizing simulated particles to employing experimental data.

To verify the possibility of this conjecture, we acquired micrograph movie stacks of the proteasome using a Titan Krios 300keV cryo-EM equipped with a K3 direct electron detection camera. The defocus range was set between 0.5µm and 1.5µm. Each stack comprised 32 frames with a total electron dose of $50e^{-1-2}$. The electron dose was uniformly distributed across all frames. Particles were picked from identical positions using averages from frames 5-14, 10-19, 15-24, and 20-29. Consequently, we constructed a dataset consisting of 183,464 particles that represented four different levels of absorbed electron doses.

This experiment demonstrates the robustness and practical applicability of CryoSieve in filtering out radiation-damaged particles. Specifically, CryoSieve is capable of sieving out the majority of particles heavily damaged by high radiation exposure, with significantly higher accuracy compared to other particle sorting algorithms such as NCC, cisTEM, AGC, and non-alignment classification.

Since experimental data may be influenced by various factors, including incorrect pose, among others, we have moved the simulated particles generated via InSilicoTEM to the supplementary section.

RC: *(4) P. 3, line 89: The authors used “high-resolution amplitude” for sorting particles in Fourier space. Have the authors sorted particles based on the high-resolution phase? It would be interesting to compare phase-based sorting with amplitude-based sorting.*

AR: We conducted an experiment where we replaced the chosen criteria in CryoSieve with the high-resolution phase residual, defined as the phase difference between the particle and the reference projection above the high-pass threshold. We utilized the TRPM8 dataset for this experiment. The high-pass threshold of each iteration and the retention ratio of each iteration were consistent with those in the TRPM8 experiment. The FSC resolution of remaining particles using high-resolution phase residual as criterion drops as iteration progresses. The results suggest that the high-resolution phase residual may not be a suitable criterion for particle sorting.

We have revised our manuscript as

Furthermore, the amplitude information within the CryoSieve score proves to be vital, given that the phase residual is ineffective as a metric for particle selection (Supplementary Figure 4).

and added Supplementary Figure 4.

RC: *(5) Page 6, lines 151-156. The authors describe the 2D and 3D classification work used in reference 26. Such a statement does not bring in new information here and should be deleted.*

AR: Thank you for your suggestion. We have deleted it.

RC: *(6-1) Page 6, lines 164-166. The authors claim that CryoSieve can remove over half of the particles with unreliable high-frequency signals without negatively affecting the final reconstruction. However, it's not clear what's the criterion/threshold to define "unreliable high-frequency signals".*

AR: Thank you for your comment. We have revised the statement as

~~These results indicate that CryoSieve can effectively eliminate over half of the particles with unreliable high-frequency signals without negatively affecting the final reconstruction. Therefore, CryoSieve is highly effective in selecting the most informative particles.~~

The results demonstrate that CryoSieve is proficient in discarding more than half of the particles, utilizing the CryoSieve score—a metric reflecting the discrepancy between the particle image and its reference projection. Crucially, this process does not compromise the quality of the final reconstruction.

RC: *(6-2) Besides, after removing the "unreliable high-frequency signals", have the authors observed improved cryo-EM densities or structural features that were blurred or missing in the published maps?*

AR: Thank you for your suggestion. Detailed comparisons of improved cryo-EM densities or structural features were plotted in Supplementary Figure 8.

We revised our manuscript as

For some datasets, the density maps showed a certain degree of improvement, which was visualized by the restoration of some previously blurred or missing side chains in the density map (Supplementary Figure 8).

RC: *(7) Page 6, lines 168-171. The authors compared CryoSieve with two other sorting methods of NCC and AGC. The authors should discuss in more detail why their method is better than the other two.*

AR: Thank you for your suggestion. The main reason that CryoSieve outperforms both NCC and AGC may be due to the incorporation of the high-pass operator in the calculation of the CryoSieve score. Through both

theoretical analysis and simulation verification, this high-pass operator exhibited superior results. We found that high-pass operator is essential in the determination of particle scores. NCC and cisTEM score calculates particle score using information across all frequencies, resulted in worse performance than CryoSieve. Without truncating high frequencies, the score is likely dominated by low-frequency components, complicating the distinction of non-contributory particles in cryo-EM.

We have revised our manuscript as

Therefore, CryoSieve significantly outperforms other particle sorting algorithms, demonstrating that the majority of particles are dispensable in the final stacks. A key factor in CryoSieve's superiority over both NCC, AGC and non-alignment classification is the integration of the highpass operator when computing the CryoSieve score. Without the truncation of high frequencies, scores may be predominantly influenced by low-frequency components, making it challenging to differentiate non-contributory particles in cryo-EM.

RC: *(8) Did the authors observe the preferred orientation issue while sorting particles based on high-frequency signals?*

AR: Thank you for the suggestion. To explore the pose distribution before and after applying CryoSieve, we visualized the directional distribution reported by CryoSPARC for all particles, the finest subset, and those particles sieved out by CryoSieve (Supplementary Figure 6). The pose distributions of the removed particles were similar to those of all particles in the final stacks.

RC: *(9) In cryo-EM, nonalignment classification is routine and effective for the classification of heterogeneous data. The nonalignment classification can sort and remove particles, in the meanwhile can identify additional conformational states. The authors should compare the performance of CryoSieve with the nonalignment classification in terms of removing particles while maintaining the resolution.*

AR: We compared the performance of CryoSieve with nonalignment classification. We applied nonalignment classification to sieve particles in the final stack. The particles were divided into four classes, and only particles belonging to the class with the highest resolution were retained. Subsequently, we used CryoSPARC to perform ab initio refinement on this selected group of particles.

For three of the eight evaluated datasets, non-alignment eliminated more than 10% of all particles, resulting in some improvement in resolution. However, this improvement is significantly less noticeable than what is observed with CryoSieve (Supplementary Material V).

We have revised this manuscript as

For the non-alignment classification applied to hemagglutinin, LAT1, and apoferritin, fewer than half of the particles were removed, resulting in some enhancement (Supplementary Material V). However, this enhancement still falls notably short of the results achieved by CryoSieve (Supplementary Material V). For the other five datasets, the retaining ratios using non-alignment classification exceeded 90%, which meant that the quality of maps reconstructed from the retained particles either remained unchanged or deteriorated (Supplementary Material V).

RC: *(10) Page 6, lines 181-183. The authors used "Einstein-from-noise" to justify the removal of the deposited Euler angles. This statement is not accurate because there is no evidence that the published reconstructions/Euler angles suffered from the "Einstein-from-noise" issue. The authors should revise the sentence to say "to remove bias in the published maps".*

AR: Thanks for your suggestion. We agree with it and have accordingly revised the statement. The main motivation for re-estimating the Euler angles is eliminating any bias in the original final stack. In this stack, the estimation of Euler angles could potentially be influenced by the particles that were discarded using CryoSieve. We revised our manuscript as

~~For all the aforementioned methods (CryoSieve, NCC, AGC, and random), we discarded the published refined Euler angles deposited on EMPIAR to avoid the Eisenstein from noise effect.~~

For all the aforementioned methods (CryoSieve, NCC, AGC, non-alignment classification, and random), we discarded the refined Euler angles published and deposited on EMPIAR to prevent the inadvertent transfer of information from the removed particles to the retained particles.

RC: (11) Page 6, lines 185-187. The authors should include all metrics, in addition to FSC and Q-score, that they have used to evaluate the maps before and after the CryoSieve procedure. As B factors are important for evaluating data quality, they should plot B factors with respect to the number of iterations in Figure 2 and Supplementary Figure 2.

AR: Thank you for your suggestions. We have plotted the Rosenthal-Henderson curve against the number of iterations for all datasets, comparing CryoSieve, NCC, cisTEM, and random, as shown in Figure 2. The content previously displayed in Supplementary Figure 2 has been incorporated into the current version of Figure 2.

RC: (12-1) Page 7, Figure 1 caption: Did the authors apply the same B-factor to sharpen the maps before and after CryoSieve?

AR: The B-factor to sharpen the maps before and after the application of CryoSieve is the same.

... the same B-factor The density maps were first FSC-weighted (based on FSCs given by CryoSPARC), and then B-factor sharpened using equivalent B-factors for the same protein, before and after CryoSieve's sieving: -90\AA^2 for TRPA1, -180\AA^2 for hemagglutinin, -100\AA^2 for LAT1, -60\AA^2 for pfCRT, -70\AA^2 for TSHR-Gs, -80\AA^2 for TRPM8, -65\AA^2 for apoferritin, and -110\AA^2 for streptavidin.

RC: (12-2) In addition to the sharpened maps, the authors may compare the non-sharpened maps in a Supplementary Figure.

AR: Thank you for the suggestion. We have added Supplementary Figure 1 to provide a comparative view of the unsharpened maps for all eight datasets, both before and after sieving.

We have revised our manuscript as

The raw density maps corresponding to these results, unsharpened by B-factor, are presented in Supplementary Figure 1.

RC: (12-3) For a better comparison, they should include the contour levels that were used to draw maps before and after CryoSieve.

AR: Thank you for your suggestion. We include the contour levels that were used to draw maps before and after CryoSieve in Figure 1 and Supplementary Figure 1.

The equivalent contour level was applied for each protein respectively, as indicated at the base of each ratio bar.

RC: (13) Page 7, lines 194-195. *If cisTEM reports a per-particle score, the authors should explain why the score can't be used as a particle sorting criterion.*

AR: We have moved the cisTEM results from the supplementary materials to the main body to facilitate a comparison with CryoSieve, NCC, AGC, and the non-alignment method, as shown in Figure 2. However, it is important to note that during the 3D refinement in cisTEM, the poses are either re-estimated or refined. This makes the comparison not strictly fair. We have highlighted this consideration in the caption of Figure 2.

We have revised our manuscript as

~~cisTEM is capable of reporting a score for each single particle image after 3D reconstruction, though it is not a particle sorting criterion. Due to differences in alignment and other image processing workflows between cisTEM and cryoSPARC, cisTEM cannot be strictly compared with CryoSieve.~~

and

CisTEM can report a score for each single particle image after 3D refinement. During the 3D refinement process of cisTEM, the pose parameters of particles are re-estimated or refined. Therefore, due to differences in alignment and other image processing workflows between cisTEM and CryoSPARC, cisTEM cannot be strictly compared with CryoSieve.

RC: (14) Page 8, lines 211-214. *Why did CryoSieve remove a substantial number of high-resolution 2D particles in TRPA1, but not in the other five data sets (Figure 2)? The authors should perform 2D class averaging on additional data sets.*

AR: Thank you for your comment. For TRPA1, the broad resolution range was not conducive for plotting, resulting in inadequate segmentation of the high-resolution range into various resolution categories in the histogram. We have added an additional histogram focusing on the partial resolution range for TRPA1, clearly indicating that particles with high resolution (7.4-7.1Å) were completely retained by CryoSieve.

We have also included two additional datasets, apoferritin and streptavidin, in Figure 2 after processing them through 2D classification. For apoferritin, particles within the highest resolution range (5.5-5.3Å) were predominantly those retained by CryoSieve. In contrast, the streptavidin dataset, possibly due to using a phase plate during data collection, displayed unusually high resolutions during the 2D classification step. This anomaly made a direct comparison between the retained and discarded particles ineffective.

We have revised our manuscript as

~~In five out of the six datasets, particle images with the highest resolution, i.e., 8.5–9.6 Å in hemagglutinin, 6.6–8.2 Å in LAT1, 7.2–11.6 Å in pfCRT, 7.2–8.5 Å in TSGH-Gs, and 11.6–7.5 Å in TRPM8, were entirely retained by CryoSieve.~~

In six out of the eight datasets, particle images with the highest resolution, i.e., 7.4–7.1 Å in TRPA1, 8.5–9.6 Å in hemagglutinin, 6.6–8.2 Å in LAT1, 7.2–11.6 Å in pfCRT, 7.2–8.5 Å in TSGH-Gs, and 11.6–7.5 Å in TRPM8, were entirely retained by CryoSieve. For apoferritin, the majority of

particles within the highest resolution range (5.5-5.3 Å) were constituted by the particles retained by CryoSieve. However, for streptavidin, possibly due to the adoption of a phase plate during data collection, unusually high resolutions were reported in the 2D classification step, rendering such a comparison between retained and removed particles ineffective.

For TRPA1 and apoferritin, the bar with the highest resolution range was further finely divided and then plotted in a histogram, which is displayed to the right of the global histogram.

RC: (15) Page 8, section 2.4: *Using simulated data, the authors claim that CryoSieve can effectively detect radiation-damaged particles better than NCC and cisTEM. The authors should also compare CryoSieve performance with the AGC method and nonalignment classification method. In addition, they should use experimental data, not just simulated data to show its effectiveness in the treatment of experimental radiation damage.*

AR: Thank you for your suggestion. We have incorporated both the ACG method and non-alignment classification in the analysis of the efficacy of removing radiation-damaged particles. We have added the experiments related to the experimental radiation damage data and compared CryoSieve performance with the AGC method and nonalignment classification method. The experimental setup is given as follows.

To verify the possibility of this conjecture, we acquired micrograph movie stacks of the proteasome using a Titan Krios 300keV cryo-EM equipped with a K3 direct electron detection camera. The defocus range was set between 0.5µm and 1.5µm. Each stack comprised 32 frames with a total electron dose of $50e^{-1-2}$. The electron dose was uniformly distributed across all frames. Particles were picked from identical positions using averages from frames 5–14, 10–19, 15–24, and 20–29. Consequently, we constructed a dataset consisting of 183,464 particles that represented four different levels of absorbed electron doses.

As shown in Figure 4, the results show CryoSieve’s proficiency in identifying particles with radiation damage compared to other particle sorting algorithms.

RC: (16) Page 9, Table 2: *For B-factor calculation, the authors should use the Rosenthal and Henderson’s B-factor method instead of values from the CryoSPARC auto-processing.*

AR: Thank you for your suggestion. We have included the Rosenthal and Henderson’s B-factors in Table 2. Additionally, the process of determining these B-factors is illustrated in Supplementary Figure 5. We also add an additional column in Figure 2, representing the Rosenthal-Henderson B-factors of CryoSieve, NCC, cisTEM, and random across iterations.

We have revised our manuscript as

The third column depicts Rosenthal-Henderson B-factors.

and

The process of fitting and solving for Rosenthal and Henderson’s B-factors is visualized in Supplementary Figure 5.

along with adding the Supplementary Figure 5.

RC: (17) Page 11, lines 330-345. *The discussion on sample preparation is off-topic to the work and should be removed or revised in the context of CryoSieve.*

AR: Thank you for your suggestion. We have removed the paragraph detailing sample preparation. It is worth noting that, in addition to progress towards theoretical limits, CryoSieve could potentially offer a quantitative metric for assessing sample quality. This could impact future technology in structural biology. Our initial collaboration with Professor Hongwei Wang supports this conjecture, and we plan to submit another report from this perspective.

RC: (18) *The authors should have shared their code as an attachment for a better evaluation of the work.*

AR: CryoSieve is now open-sourced and available on GitHub. A detailed tutorial can also be found on its homepage. Additionally, the datasets used in this manuscript, along with the expected outputs after running CryoSieve, have been deposited on GitHub and can be accessed via CryoSieve's homepage.

We have updated the 'Code Availability' section in this manuscript accordingly.

~~CryoSieve will be open-source upon publication and is also available upon request during the review process.~~

CryoSieve is now open-sourced and available on GitHub (<https://github.com/mxhulab/cryosieve>). A detailed tutorial can also be found on its homepage. Moreover, datasets used in this manuscript, along with the expected outputs after running CryoSieve, have been deposited on GitHub and can be accessed via CryoSieve's homepage.

Reviewers' Comments:

Reviewer #1:

Remarks to the Author:

The revisions for the manuscript "Not final yet: a minority of final stacks yields superior amplitude in single-particle cryo-EM" from Zhu et al, which approaches the important question in cryo-EM of the number of images required to get high resolution structures, greatly improved the quality of the paper. The additional experiments comparing the results of the method for CryoSPARC and RELION are quite intriguing. The additional supplemental figures, description of the method, and release of the github code are an excellent addition.

Reviewer #2:

Remarks to the Author:

The revised manuscript by Zhu, et. al has been much improved. All of my concerns were mostly addressed satisfactorily. Here are a few minor things for authors to consider to further improve the accuracy and readability of their paper.

P. 3, lines 101-103. "We conclude that, for these datasets, the opportunity for further improvement lies in generating fewer futile particles during sample preparation rather than further improving the quality of the particle images that constitute the finest subset.". This sentence may lead to the interpretation that the images in the finest subset is perfect which is true. Please revise the sentence.

P. 4. Line 123. "It tends to deviate significantly from the true amplitude." How did the authors obtain the true amplitude? Which module did the authors use in their cryoSPARC reconstructions, homologous refinement or reconstruction only? The authors may need to use the correct module in cryoSPARC as they did for Relion (relion_reconstruct) to get comparable results.

P.7. Lines 200-202. "For the non-alignment classification applied to hemagglutinin, LAT1, and apoferritin, less than half of the particles were removed, resulting in some enhancement (Supplementary Material V)." For non-alignment classification, one can remove more particles by varying the number of classes (K), the regularization parameter (tau2_fudge), and the number of iterations. In general, a higher tau2_fudge value (>4) may lead to the removal of more particles. How many iterations did they use nonalignment for their classifications? The authors might want to play with these parameters and include them in their Supplementary Material V.

P.7. Lines 213-215. "Thus, the retained particles were used for ab initio reconstruction by CryoSPARC to obtain refreshed sets of Euler angles and density maps." This sentence causes a confusion with their previous argument that "CryoSPARC is not the optimal choice for reconstruction." (P. 4, line 122). The authors should clarify how they use cryoSPARC and Relion for each of the steps in order to perform CryoSieve. Adding external programs and modules to Supplemental Figure 3 would be helpful.

P. 21. Fig. 2. The y-axis label of B factor plots in the right panels shouldn't be reverted. B-factors can be plotted the same as the left panels for resolution, with a higher resolution, i.e. smaller value, at the top of the y-axis.

P. 23. Fig. 4. It's strange that panel a starts from the second row. On the PDF version, I couldn't tell any particles on the images. The authors should rearrange their panels and consider replacing panel a with a different plot or just a diagram to show how they distributed the total dose across four subsets. In addition, the standard format of dose is $e^{-1}/\text{\AA}^2$, not $e^{-1}/\text{\AA}^2$.

Authors' Response to Reviews of

A minority of final stacks yields superior amplitude in single-particle cryo-EM

Jianying Zhu, Qi Zhang, Hui Zhang, Zuoqiang Shi, Mingxu Hu and Chenglong Bao
Submitted to *Nature Communications*, NCOMMS-23-22170A

RC: *Reviewers' Comment*, **AR:** Authors' Response, Manuscript Text

We sincerely thank the valuable suggestions and comments from the reviewers. We list our point-to-point replies in the following context and hope that the revision can address the concerns.

Response to Referee #1

RC: *The revisions for the manuscript "Not final yet: a minority of final stacks yields superior amplitude in single-particle cryo-EM" from Zhu et al, which approaches the important question in cryo-EM of the number of images required to get high resolution structures, greatly improved the quality of the paper. The additional experiments comparing the results of the method for CryoSPARC and RELION are quite intriguing. The additional supplementary figures, description of the method, and release of the github code are an excellent addition.*

AR: Thanks for your support of this paper.

Response to Referee #2

RC: (1) *The revised manuscript by Zhu, et. al has been much improved. All of my concerns were mostly addressed satisfactorily. Here are a few minor things for authors to consider to further improve the accuracy and readability of their paper.*

AR: Thanks for your support. We appreciate your suggestions for improving our paper.

RC: (2) *P. 3, lines 101-103. “We conclude that, for these datasets, the opportunity for further improvement lies in generating fewer futile particles during sample preparation rather than further improving the quality of the particle images that constitute the finest subset.” This sentence may lead to the interpretation that the images in the finest subset is perfect which is true. Please revise the sentence.*

AR: Thank you for your comment. We have revised the statement as

From our experiments, we suggest that advancements during the sample preparation process, aimed at increasing the proportion of the finest subset in the final stack, could potentially facilitate the development of cryoEM.

~~We conclude that, for these datasets, there is greater room for further improvement in generating fewer futile particles during sample preparation than further improving the quality of the particle images that constitute the finest subset.~~

RC: (3) *P. 4. Line 123. “It tends to deviate significantly from the true amplitude.” How did the authors obtain the true amplitude? Which module did the authors use in their cryoSPARC reconstructions, homogeneous refinement or reconstruction only? The authors may need to use the correct module in cryoSPARC as they did for Relion (relion_reconstruct) to get comparable results.*

AR: In our experiment, we utilized the reconstruction-only module in CryoSPARC reconstructions, which is understood to perform a similar function to relion_reconstruct. Given that RELION is open-source software deploying Fourier central-slice-theorem-based methods for reconstruction, we can verify its output of true amplitudes. In contrast, our findings indicate that the output amplitudes from CryoSPARC differ from those from RELION in terms of magnitude by several orders. Furthermore, the exact reconstruction process of CryoSPARC remains unclear to us due to its closed-source nature. For these reasons, we have found that the density reconstructed by CryoSPARC significantly deviates from the true amplitude.

RC: (4) *P.7. Lines 200-202. “For the non-alignment classification applied to hemagglutinin, LAT1, and apoferritin, less than half of the particles were removed, resulting in some enhancement (Supplementary Material V).” For non-alignment classification, one can remove more particles by varying the number of classes (K), the regularization parameter (tau2_fudge), and the number of iterations. In general, a higher tau2_fudge value (>4) may lead to the removal of more particles. How many interactions did they use nonalignment for their classifications? The authors might want to play with these parameters and include them in their Supplementary Material V.*

AR: Thank you for your suggestion. We varied the number of classes (K), the regularization parameter (tau2_fudge) and the number of iterations for experimental datasets. The particles were divided into K classes, and only particles belonging to the class with the highest resolution were retained. The number of classes (K) is 20 or 40, the number of iterations (iter_num) is 40 or 80, and tau2_fudge is 6 or 8. We selected the particle with the highest FSC resolution of the reconstruction density map from these eight possible combinations and reported them. The box size of TSHR-Gs was 448, the particle number of LAT1 and apoferritin were 250,712

and 382,391, our machine run out of memory for some parameters of these datasets, so the iteration number of TSHR-Gs, LAT1, and apoferritin were set to 25.

The additional experimental results were given in Supplementary Material V (Supplementary Table 5 and Supplementary Table 6).

In addition, we conducted variations in several parameters across all eight experimental datasets: the number of classes (K), the regularization parameter (tau2_fudge), and the number of iterations used in nonalignment classification. In this process, particles were categorized into various classes, with only those in the highest-resolution class being retained for further analysis. Specifically, we tested the configurations where the number of classes (K) were set to 20 and 40, the number of iterations (iter_num) were 40 and 80, and tau2_fudge values were chosen as 6 and 8. We selected the particle with the highest FSC resolution of the reconstruction density map from these eight possible combinations of parameters and reported them. The box size of TSHR-Gs was 448, the particle number of LAT1 and apoferritin were 250,712 and 382,391, our machine run out of memory for some parameters of these datasets, so the iteration number of TSHR-Gs, LAT1, and apoferritin were set to 25. Subsequently, we used CryoSPARC to perform ab initio refinement on this selected group of particles. The results are shown in Supplementary Table 5 and Supplementary Table 6

RC: (5) P.7. Lines 213-215. *“Thus, the retained particles were used for ab initio reconstruction by CryoSPARC to obtain refreshed sets of Euler angles and density maps.” This sentence causes a confusion with their previous argument that “CryoSPARC is not the optimal choice for reconstruction.” (P. 4, line 122). The authors should clarify how they use CryoSPARC and Relion for each of the steps in order to perform CryoSieve. Adding external programs and modules to Supplementary Figure 3 would be helpful.*

AR: We used RELION to reconstruct density maps from final stacks in the particle selection step, and then used CryoSPARC to re-estimate poses of the retained particles in ab initio refinement step.

~~Given that g_j relies on the accurate amplitude of the reconstructed density map $x^{(k)}$, CryoSPARC is not the optimal choice for reconstruction (Supplementary Figure 2).~~

Given that g_j relies on the accurate amplitude of the reconstructed density map $x^{(k)}$, CryoSPARC is not the optimal choice for reconstruction in particle selection step (Supplementary Figure 2).

We have modified the flow chart in Supplementary Figure 3, including the modules and software we used.

RC: (6) P. 21. Fig. 2. *The y-axis label of B factor plots in the right panels shouldn't be reverted. B-factors can be plotted the same as the left panels for resolution, with a higher resolution, i.e. smaller value, at the top of the y-axis.*

AR: We appreciate your suggestion, and Figure 2 has been revised accordingly.

RC: (7) P. 23. Fig. 4. *It's strange that panel a starts from the second row. On the PDF version, I couldn't tell any particles on the images. The authors should rearrange their panels and consider replacing panel a with a different plot or just a diagram to show how they distributed the total dose across four subsets. In addition, the standard format of dose is $e^{-1/\text{Å}^2}$, not $e^{-1/\text{Å}^2}$.*

AR: Thank you for your suggestion. We have re-arranged the layout of panels in Figure 2, placing panel (a) in the 1st column of this figure. Additionally, we have corrected the unit from $e^{-1/\text{Å}^2}$ to $e^{-\text{Å}^{-2}}$.

Figure 1: **Flow chart scheme for CryoSieve.** CryoSieve operates through multiple iterations. Each iteration comprises both density map reconstruction and particle sieving.

Given the extremely low signal-to-noise ratio (SNR) in single-particle cryo-EM images, directly observing particles within images is difficult. Panel (a) illustrates how variations in electron dose do not result in noticeable changes when observed with the naked eye, thereby highlighting the need for a specialized algorithm for analysis. In response to this requirement, we propose the utilization of CryoSieve.